# The Butterfly Effect: Neural Network Training Trajectories Are Highly Sensitive to Initial Conditions

**Gül Sena Altıntaş** [* 1 2] **Devin Kwok** [* 3 4] **Colin Raffel** [1 2] **David Rolnick** [3 4]

## Abstract

Neural network training is inherently sensitive to initialization and the randomness induced by stochastic gradient descent. However, it is unclear to what extent such effects lead to meaningfully different networks, either in terms of the models' weights or the underlying functions that were learned. In this work, we show that during the initial "chaotic" phase of training, even extremely small perturbations reliably causes otherwise identical training trajectories to diverge—an effect that diminishes rapidly over training time. We quantify this divergence through (i) $L^2$ distance between parameters, (ii) the loss barrier when interpolating between networks, (iii) $L^2$ and barrier between parameters after permutation alignment, and (iv) representational similarity between intermediate activations; revealing how perturbations across different hyperparameter or fine-tuning settings drive training trajectories toward distinct loss minima. Our findings provide insights into neural network training stability, with practical implications for fine-tuning, model merging, and diversity of model ensembles. [1]

## 1. Introduction

Neural network training is known to be unstable in the sense that noise can disrupt convergence to a particular minimum (Frankle et al., 2020a; Wu et al., 2018). This is true even when considering solutions that perform equally well, since symmetries and connected minima in the loss landscape give rise to many different ways for a neural network to

---
[*]Equal contribution [1]University of Toronto [2]Vector Institute [3]McGill University [4]Mila – Quebec AI Institute. Correspondence to: Gül Sena Altıntaş <gsaltintas@cs.toronto.edu>, Devin Kwok <devin.kwok@mail.mcgill.ca>.

*Proceedings of the $42^{nd}$ International Conference on Machine Learning*, Vancouver, Canada. PMLR 267, 2025. Copyright 2025 by the author(s).

[1]Our code is available at https://github.com/gsaltintas/lmc

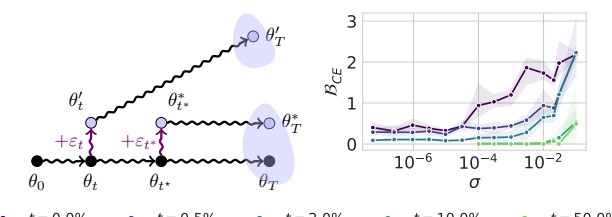

*Figure 1.* **Left**: illustration of the "butterfly effect": a network $\theta_0$ is trained until time $t$ and perturbed by $\varepsilon$ early ($\varepsilon_t$) or later ($\varepsilon_{t*}$) in training. Both copies are trained deterministically until $T$ and their divergence is measured (purple loss basins). **Right**: barriers (training cross-entropy loss) at $T$ versus perturbation magnitude ($\sigma = 1$ is the network's initialization scale). A perturbation of *as little as one weight* (leftmost points) reliably causes divergence when applied early, but not when applied at later $t$ (colors).

parameterize identical or similar functions.

Although training instability affects convergence in general (Iyer et al., 2023; Jastrzebski et al., 2020), it also prevents different runs of the same network from consistently reaching one particular solution, which has practical implications for model merging (Singh & Jaggi, 2020; Ainsworth et al., 2023) and ensembling. Prior work has categorized training into chaotic (early) and stable (late) phases (Fort et al., 2020; Frankle et al., 2020a), but it is not clear if instability is more a product of noise (e.g. batch noise, data augmentations, GPU indeterminacy), a network's current state (e.g. random vs. pre-trained), or the training procedure itself (e.g. optimizer and hyperparameter selection). A thorough understanding of instability should disentangle these factors, since their influence may vary over training time and between different settings. Furthermore, depending on the task, not all of these factors can be controlled— when fine-tuning a pre-trained model one can vary hyperparameters but not the initial weights, for instance, while the opposite may be true when pre-training the same model.

These limitations motivate us to study training stability absent the effects of noise. Drawing from the dynamical systems perspective, we consider how much a *deterministic* training map diverges when a controlled perturbation is applied to its initial conditions (i.e. the starting weights of a network). By selecting initial weights from models trained

or fine-tuned for varying durations, we build up a picture of where in the loss landscape training trajectories tend to diverge, and how sensitive trajectories are to perturbations in these regions. Crucially, our approach can quantify stability more precisely and for a wider range of models—from randomly initialized to pre-trained—than was possible in prior works, which only measured instability to training noise. Our contributions are as follows:

1. We show that a tiny perturbation of *as little as a single weight* early in training causes two otherwise identically initialized and trained networks to diverge—the *butterfly effect* (Figure 2).

2. Conversely, even networks that are stable to training noise diverge under larger perturbations. This points towards the possibility of using perturbations during training or fine-tuning to increase model diversity and ensembling performance (Figure 3).

3. We find that stability improves under settings including wider/shallower networks and increased learning rate warm-up (among others), and these settings can be combined to further decrease, but not eliminate, instability near initialization (Figure 4).

4. While pre-trained networks are orders of magnitude more stable than randomly initialized networks, stability varies greatly between tasks and remarkably, more pre-training of language models can actually *reduce* stability in some cases (Figure 5).

5. Contrary to a dynamical systems perspective, $L^2$ and barriers (Frankle & Carbin, 2019) do not grow exponentially over training (Figure 6), and although $L^2$ and barriers are strongly correlated in some cases, this is not true generally (Figure 7).

## 2. Related Work

*Stability and optimization.* Many works have studied the stability of optimization by asking if neural networks converge to well-generalizing minima (Cohen et al., 2021; Jastrzebski et al., 2020; Wu et al., 2018) or converge at all (Iyer et al., 2023; Jacot et al., 2018; Sohl-Dickstein, 2024). However, these works do not consider stability relative to any particular training trajectory.

We focus on the question of whether training from an initial point tends to follow the same trajectory to a specific "loss basin", i.e. a linearly connected, low-loss region of the loss landscape. Although narrower in scope, this question is highly relevant for practical contexts such as when using pre-trained models or conducting fine-tuning. In order to merge or ensemble models in these contexts, one may not merely want to find a good solution, but may, for example, want to converge towards or away from a particular pre-existing solution in order to improve merge-ability or model diversity, respectively.

*Dynamical system stability.* Neural network training has been studied as a stochastic process (Smith & Le, 2018; Teh et al., 2016; Redman et al., 2024) and as a dynamical system (Wu et al., 2018; Jastrzebski et al., 2020; Cohen et al., 2021). We take the latter view and analyze how small perturbations to a network's initial weights evolve over training. This has two advantages: first, we can differentiate between instability due to noise vs. instability inherent to training itself (in the same way that deterministic dynamical systems exhibit chaos); and second, we can model simpler and larger (exponential) instabilities, which if present, should dominate over stochastic effects and make the stochastic perspective unnecessary (Wu et al., 2018).

Prior works (Wu et al., 2018; Jastrzebski et al., 2020; Cohen et al., 2021) have considered dynamical systems stability in terms of whether the $L^2$ distance between a network's weights and a fixed minimum will grow over time. Our work differs in that we evaluate stability relative to a moving trajectory, and we also want to know if networks are diverging in function (not only in weights). To measure functional divergence, we use barriers (Frankle et al., 2020a), barriers accounting for neural network permutation symmetries (Entezari et al., 2022; Ainsworth et al., 2023), and representational similarity via Angular CKA (Williams et al., 2021). Although these quantities are not amenable to approximation by linear dynamical systems, they better capture the practical differences between networks.

*Linear mode connectivity.* Barriers (Frankle et al., 2020a; Neyshabur et al., 2020) are the maximum increase in loss on a linear path between two networks. Networks with barriers below some noise threshold are said to exhibit *linear mode connectivity* (LMC), which among other useful properties is a necessary condition for the loss landscape to be locally convex (Neyshabur et al., 2020, Definition 3.1). Thus, non-zero barriers indicate that two networks belong to different convex loss basins (Goodfellow et al., 2015; Huang et al., 2017; Yunis et al., 2022).[2]

Note that we do not consider the more general notion of (non-linear) mode connectivity for several reasons. First, despite being non-convex, neural network training is often understood in terms of convex optimization. In convex regions of the loss landscape however, convex optimization behaves exactly as expected. Similarly, the dynamical systems perspective often takes a linear or quadratic approximation of the loss landscape, which again holds precisely in convex regions. Practically, merging models by weight averaging requires linearly connected networks, which is guaranteed if

---

[2]Following Entezari et al. (2022); Neyshabur et al. (2020), we make the extra assumptions needed to assume the converse, i.e. that networks with zero barrier are in the same loss basin.

the networks are from the same convex region. Finally, both theoretical Simsek et al. (2021); Lin et al. (2024) and empirical (Draxler et al., 2018; Garipov et al., 2018; Sonthalia et al., 2024) works have suggested that most or all minima may be trivially connected by non-linear paths.

*Spawning experiments.* Frankle et al. (2020a;b) and Fort et al. (2020) consider if training is stable to random batch order and data augmentations (training noise) by spawning pairs of networks from the same parent, and measuring barriers between them after training. They find that training becomes stable to training noise after an early period of instability. Altıntaş et al. (2023) shows that reducing early training variability by lowering learning rates, increasing batch sizes, and adding learning rate warm-up further increase stability to training noise, and Singh et al. (2024) relates stability in barriers to the loss landscape geometry. These findings align with observations that after an initial chaotic phase, SGD training trajectories can be approximated by a linear kernel (Fort et al., 2020).

We adopt the same parent-child spawning experiments (Frankle et al., 2020a;b; Fort et al., 2020), but we eliminate all training noise so as to isolate the effects of perturbation to specific training times. This not only lets us precisely identify when and how much perturbation causes instability, but also allows us to do so on both randomly initialized (chaotic) and pre-trained (stable) networks.

*Model averaging.* Weight averaging can merge zero-barrier networks from the same training trajectory (Izmailov et al., 2018), different runs (Utans, 1996; Wortsman et al., 2021), or different tasks (Mirzadeh et al., 2021) to improve inference speed and even performance. Weight averaging is the basis for more sophisticated model merging strategies (Ilharco et al., 2023), including those using permutations to align diverging (Wang et al., 2020) or unrelated (Singh & Jaggi, 2020) models. Our work indicates the conditions in which models are stable with respect to barriers, and thus amenable to weight averaging.

*Permutation symmetries.* Recent works have shown that independently initialized networks can converge to linearly connected basins after accounting for permutation symmetries, or different ways to order the neurons in a network (Entezari et al., 2022; Ainsworth et al., 2023; Benzing et al., 2022). The permutations aligning two networks are most unstable early in training (Sharma et al., 2024), suggesting a connection with training instability. While our work compares identical rather than randomly initialized networks, we apply weight and activation matching algorithms in the same manner as Ainsworth et al. (2023) to determine (1) if training instability causes permutations between networks, and (2) if undoing these permutations returns diverging trajectories to the same loss basin.

*Representational similarity.* Representational similarity compares the intermediate (hidden) outputs of two networks, and can detect functional differences even when two networks have identical performance. Although many methods exist, we use Angular CKA (Williams et al., 2021) which we explain and justify in detail in Appendix B.2. In general, representational similarity methods are invariant to symmetries of a network's outputs, but not of its weights. Thus, dissimilar representations indicate greater diversity between networks (which can improve ensembling performance), but similar representations do not guarantee that two networks are in the same loss basin (due to weight symmetries), and thus is not sufficient for weight averaging to succeed.

*Fine-tuning stability.* Pre-trained models are generally stable to training noise and converge to the same basin during fine-tuning (Neyshabur et al., 2020). However, more recent work has found that this is not always true, as Juneja et al. (2023) discovered that training noise causes fine-tuning of language models to converge to different basins. While detrimental to model merging, this kind of instability can improve model diversity and thus ensemble performance (Lubana et al., 2023; Sadrtdinov et al., 2023), even if a single basin has equivalent diversity (Lion et al., 2024).

Our method enables us to find the threshold between stability and instability for any fine-tuning setting—even ones stable to training noise—since we can increase our perturbations to any scale necessary for inducing instability in a given network. Using our method, we identify differences in stability between language and vision models, specifically studying the fine-tuning dynamics of ResNets (He et al., 2016), ViT (Dosovitskiy et al., 2021), BERT (Devlin et al., 2019), and OLMo (Groeneveld et al., 2024).

## 3. Methods

In this section, we define our notion of training stability, describe the framing for our perturbation experiments, and finally, define the functional dissimilarity scores and other quantities we evaluate (barriers, barriers modulo permutation, and $L^2$ divergence over training time).

### 3.1. Training Instability

Consider training as the iterative application of a stochastic training map $\mathcal{T} : \Theta \to \Theta$ to the initial parameters $\theta_0 \in \Theta$ of a neural network, so that the network's parameters after training are

$$\theta_T = \mathcal{T}^T(\theta_0; \xi) = \underbrace{\mathcal{T} \circ \mathcal{T} \circ ... \circ \mathcal{T}}_{T \text{ times}}(\theta_0; \xi_1, ..., \xi_T), \quad (1)$$

where $\xi = (\xi_1, \ldots, \xi_T)$ accounts for all of the stochastic factors influencing training, such as batch sampling, data augmentation, and hardware-induced non-determinism. Un-

less specified, we treat $\theta$ as a vector concatenation of a network's parameters, and when writing $\mathcal{T}$ we omit the training or test data if it can be inferred from context.

We are interested in the degree to which training is stable, in the sense that a small perturbation to a network's initial weights does not significantly change the network after training. To describe stability on a continuum, we evaluate how far $\theta_T$ and $\theta'_T$ have diverged after training according to various notions of similarity (Section 3.4). We choose $T$ so that networks converge to a similar level of training and test performance (see details in Appendix A).

If we fix a particular $\xi$, $\mathcal{T}^T$ describes a dynamical system whose outcome depends only on the initial parameters $\theta_0$. This perspective has numerous advantages. First, we can separate the effects of training noise and isolate instability to the action of $\mathcal{T}$. As dynamical systems can diverge at exponential rates, dominating over stochastic effects (Wu et al., 2018), our deterministic experiments also lower bound the instability of regular stochastic training—i.e.

$$\mathbb{E}\left[d\left(\mathcal{T}(\theta,\xi),\mathcal{T}(\theta+\varepsilon,\xi')\right)\right] \geq \mathbb{E}\left[d\left(\mathcal{T}(\theta,\xi),\mathcal{T}(\theta+\varepsilon,\xi)\right)\right]$$

for independently sampled noise $\xi$ and $\xi'$ and a similarity measure $d$.

### 3.2. Spawn-And-Perturb Experiment

Our experiment adapts the parent-child spawning experiment introduced by Frankle et al. (2020a) to the notion of stability introduced above. The procedure is as follows:

1. Choose an initial state $\theta_0$ for a network.

2. Train the network until the perturbation time $t$, giving $\theta_t = \mathcal{T}^t(\theta_0\,;\,\xi_{1:t})$.

3. Make two copies of the network, and perturb one by adding $\varepsilon$ noise with magnitude $\sigma$ to get $\theta'_t = \theta_t + \sigma\varepsilon$.[3]

4. Train both original ($\theta_t$) and perturbed ($\theta'_t$) copies with identical training noise to get $\theta_T = \mathcal{T}^{T-t}(\theta_t\,;\,\xi_{t:T})$ and $\theta'_T = \mathcal{T}^{T-t}(\theta'_t\,;\,\xi_{t:T})$.

5. Measure the resulting instability via $d\left(\theta_T, \theta'_T\right)$, where $d: \Theta \to \mathbb{R}^+$ is a dissimilarity score.

By controlling $\theta_0$ and $t$, we can explore the stability of different points $\theta_t$ in the loss landscape. More specifically, we select between different trajectories by randomly initializing $\theta_0$ or setting it to a pre-trained checkpoint from another task. We then vary the perturbation time $t$ to examine how stability evolves during training. Changing $\mathcal{T}$ (by choosing different model architectures, optimizers, hyperparameters,

---

[3] For interpretability, $\|\varepsilon\|_2^2$ is normalized to match the expected scale of the network at initialization—see Appendix B.3 for details.

or training tasks) enables comparisons between different loss landscapes.

To quantify instability, we record the rate at which a dissimilarity score $d(\theta_T, \theta'_T)$ increases relative to the perturbation magnitude $\sigma$. By sampling perturbations of different sizes and directions, we can estimate the size and shape of the local region around $\theta_t$ where $\mathcal{T}$ does not tend to cause divergence in terms of $d$.

Our experiment differs from Frankle et al. (2020a) and Fort et al. (2020) in that they use independent training noise starting at $t$, instead of a single perturbation, to induce instability. While this reflects the stability of ordinary training, our experiments have two key advantages: we can isolate our instability analysis to specific parts of the training trajectory from $t$ to $T$, and we can also apply much smaller or larger perturbations than training noise to quantify instability over a broader scale. In Figure 10, we verify that instability in our method—i.e. to single perturbations—implies instability in the methods of Frankle et al. (2020a); Fort et al. (2020)—i.e. to training noise.

### 3.3. Perturbations

The stability of a dynamical system around a given point is direction-dependent. We take this into consideration by sampling perturbations using two different methods, which give either a narrow or broad distribution of directions. As described in Appendix B.3, all noise samples are also normalized to a fixed $L^2$ relative to the network's initialization scale.

**Batch perturbation.** Batch perturbations (Eq. 2) measure stability along the directions most likely to be explored during training, by simulating a single independently sampled optimization step:

$$\hat{\varepsilon}_{\text{Batch}} = \frac{1}{n}\sum_{i=1}^{b}\nabla\ell(x_i, y_i; \theta_t), \qquad x_i, y_i \sim \mathcal{D}, \quad (2)$$

where $\nabla\ell$ is the gradient of the loss function $\theta_t$ the network weights, and $(x_i, y_i)$ are $b$ examples sampled from a minibatch of the training dataset $\mathcal{D}$. Ignoring factors like momentum, this is equivalent to taking an extra training step at time $t$, which is rescaled by the perturbation magnitude $\sigma$ instead of the learning rate.

**Gaussian perturbation.** To measure stability in all directions generally, we use Gaussian perturbations (Eq. 3), which are scaled versions of the the network's distribution at random initialization. For networks initialized with a Kaiming/He normal distribution (He et al., 2015), Gaussian

perturbations are sampled as

$$\hat{\varepsilon}_{\text{Gaussian}} = \left[ \varepsilon_i^{(l)} \right], \qquad \varepsilon_i^{(l)} \sim \mathcal{N}\left( 0, \frac{2}{n_{l-1}} \right) \quad (3)$$

where $\varepsilon_i^{(l)}$ is the perturbation for the $i$th weight of layer $l$, $n_{l-1}$ is the number of inputs from the preceding layer (commonly called *fan-in*), and $\mathcal{N}(0, s)$ is the normal distribution with mean 0 and standard deviation $s$.

Although this is not strictly uniform in all directions (when considering all of a network's weights as a single vector), we choose to match the scale of the network's initialization to ensure that perturbations do not disproportionally affect some layers more than others. Since biases and normalization weights have constant initialization, we do not perturb them in our main experiments.[4]

### 3.4. Evaluating Functional Similarity

We use four methods to evaluate the functional similarity of networks in our spawn-and-perturb experiment: (1) $L^2$ distance in weight space $\|\theta_T - \theta_T'\|_2$, (2) the loss barrier in Equation (4), (3) the loss barrier after accounting for permutation symmetries using the weight matching algorithm from (Ainsworth et al., 2023), and (4) the representational similarity of intermediate layers measured via Angular CKA (Williams et al., 2021).

$L^2$ **divergence.** In a linear dynamical system, the $L^2$ divergence $\|\theta_T - \theta_T'\|_2$ can diverge exponentially over time at a rate of $\|\varepsilon\|_2 e^{\lambda t}$, where $\lambda$ is curvature dependent (see Appendix B.4 for derivation). To determine whether this linear approximation holds for neural network training, we measure $L^2$ distance between parameter vectors over the course of training to look for exponential growth and to determine whether $L^2$ divergences are proportional to the perturbation magnitude $\sigma$.

**Barriers.** Barriers measure the maximum increase in loss or error along the linear path between weights (Frankle et al., 2020a; Neyshabur et al., 2020). We measure the training loss barrier as

$$\sup_{\alpha \in (0,1)} \ell(\alpha\theta_T + (1-\alpha)\theta_T') - \alpha\ell(\theta_T) - (1-\alpha)\ell(\theta_T'), \quad (4)$$

where $\alpha$ interpolates between the networks and $\ell$ is the loss function for the training data. Since our work is concerned with the shape of the loss landscape in which training occurs, we report the cross-entropy loss barrier for training data ($\mathcal{B}_{\text{ce}}$) and after accounting for permutations ($\mathcal{B}_{\text{ce}}^{WM}$) throughout the main text. Appendix B.1 describes how we compute barriers in more detail.

---

[4]Figure 9 shows results from perturbing biases and normalization weights.

**Barriers modulo permutation.** To consider whether training instability causes networks to converge to different permuted versions of the same loss basin, we apply the weight and activation matching algorithms from Ainsworth et al. (2023) to find a permutation of neurons $P$ that approximately minimizes the $L^2$ distance between two networks' weights or intermediate activations, respectively. To measure the degree to which $\theta_T$ and $\theta_T'$ have been permuted with respect to each other, we record both the barrier between $\theta_T$ and $P[\theta_T']$, and the fraction of identity elements (unpermuted neurons) in $P$.

As the two matching algorithms may not necessarily find the permutation that best minimizes barriers (i.e. the *barrier modulo permutation*), we follow Sharma et al. (2024) in treating the barrier between $\theta_T$ and $P[\theta_T']$ as an upper bound. As a result, if the barrier after permuting by $P$ is significantly reduced, we can say that training instability mainly causes permutations that do not substantially change a network's function. However, if $P$ does not reduce the barrier between $\theta_T$ and $\theta_T'$, this could either mean that training instability causes networks to learn different functions, or that we have merely failed to find a permutation that does reduce barriers.

**Representational similarity.** We also consider whether networks in our experiments differ in their penultimate representations using the angular version of Centered Kernel Alignment (CKA), a type of representational similarity metric. CKA measures the cross-correlation between two arbitrary representations of the same data (Kornblith et al., 2019). As defined in Equation (5), we use Angular CKA with a linear kernel (Williams et al., 2021) to measure the distance between the outputs of the last residual or attention block. Since the resulting distance is an angle, 0 indicates that two networks have perfectly similar representations, and $\pi/2$ indicates that two networks have dissimilar representations. Appendix B.2 includes full details.

## 4. Experiments

Full training details, including hyperparameters and train/test performance, are listed in Appendix A.

### 4.1. Early vs. Late Training Instability

We first train residual networks (He et al., 2015) on CIFAR-10 (Krizhevsky, 2009) with SGD, using standard data augmentations and hyperparameter settings, including a 2% warm-up period followed by linearly decaying learning rate.

**Early perturbations reliably cause large barriers.** Figure 2 shows that training from randomly initialized networks is highly sensitive to initial conditions, as batch perturbations as small as 0.01% of a network's weights produce large barriers. We further reduce the perturbation magnitude

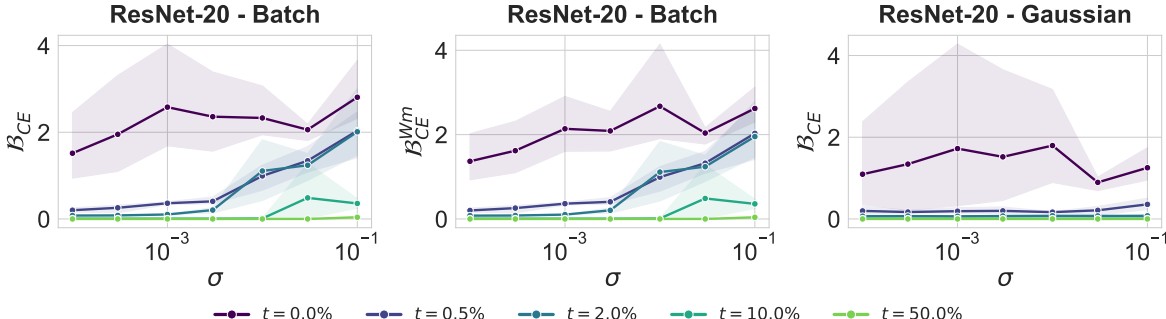

*Figure 2.* Stability of ResNet-20 trained on CIFAR-10 with SGD (details in Appendix A). Loss barriers on training data at the end of training (y-axis) are plotted against perturbation magnitude (x-axis) and perturbation step (color indicates fraction of total training time). **Left:** barriers due to batch perturbation. **Middle:** batch perturbation barriers after accounting for permutations. **Right:** barriers due to Gaussian perturbation. For the same plots with log-scaled y-axes, see Figure 27.

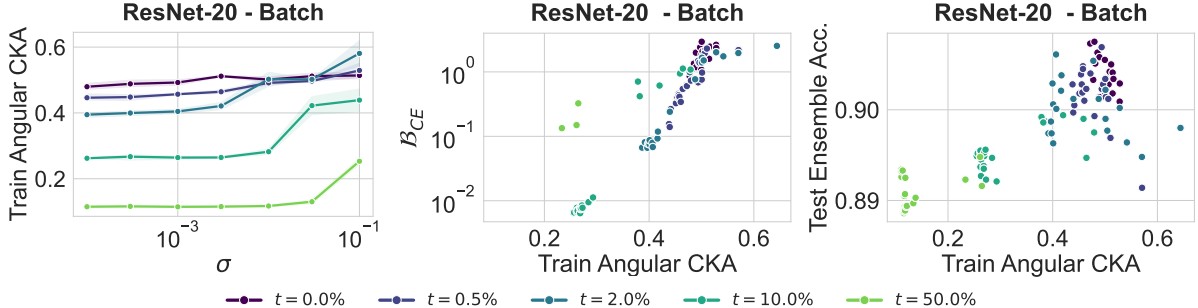

*Figure 3.* **Left:** same as Figure 1 but measuring representational similarity distance via Angular CKA (y-axis), defined in Equation (5), between original and perturbed models after training for various perturbation times (colors) and magnitudes (x-axis). **Middle:** barriers versus Angular CKA. **Right:** test accuracy of an ensemble of the original and perturbed models after training (averaging logits), versus Angular CKA. See Figures 13 to 15, 21 and 23 for more hyperparameter settings and fine-tuning of ViT and BERT, respectively.

by modifying only a fraction of the weights, finding that a single perturbed weight is sufficient to cause instability (Figure 11).

While prior work has shown that training noise near initialization causes barriers (Fort et al., 2020; Frankle et al., 2020a), we are the first to show that barriers can occur with extremely small perturbations concentrated in the first few steps, as applying the same perturbations as early as 0.5% of the way through training results in significantly reduced barriers. We name this initial instability after the "butterfly effect" in chaotic dynamical systems.

The stability increases over the first 0.5% of training time is well within the 2% warm-up period we use, and only very large perturbations (10% of initialization) result in non-zero barrier after 50% of training time. While prior works find that models become stable to training noise after the first few epochs of training (Fort et al., 2020; Frankle et al., 2020a), we quantify the scale of perturbation needed to induce barriers beyond this critical point, showing that stability continues to increase throughout training.

**Early instability is direction-independent.** Comparing batch versus Gaussian perturbations (Figure 2), we find that although networks are more stable to the latter (which are evenly distributed in direction), networks perturbed at initialization have high barriers for both. This shows early instability is mainly attributable to the network's state, and not the direction or magnitude of perturbation. However, later instability does vary depending on the direction of perturbation, which suggests that findings that use training noise (Fort et al., 2020; Frankle et al., 2020a) may not be transferrable to other kinds of perturbations.

**Training divergence is unlikely to be caused by permutations.** Comparing barriers with and without permutation alignment (Figure 2), we find that applying permutations to minimize the $L^2$ distance between networks does not reduce barriers.[5] While we cannot rule out the possibility that better (and more costly) alignment methods may reduce barriers, we argue that this is unlikely. Prior work aligning

---

[5] We omit $L^2$ distance between networks after alignment, as it is generally not reduced greatly by weight matching (Ito et al., 2025).

*differently* initialized networks finds that weight matching can reduce barriers to some degree even when it is outperformed by other methods (Peña et al., 2023; Navon et al., 2024; Ainsworth et al., 2023), whereas in our case of *identically* initialized networks, weight matching is unable to reduce barriers at all. This suggests that training instability produces real functional differences between networks, as opposed to simply permuting weights that are otherwise equivalent.

## 4.2. Functional Diversity

Comparing the similarity of intermediate representations in Figures 3 and 23, we find that Angular CKA (Eq. 5) correlates with earlier and larger perturbations (left), as well as barriers after training (middle). This again indicates functional differences beyond weight symmetries.[6]

Since model ensembling benefits from diversity, we also consider whether intentionally perturbing networks can improve ensembling performance. This effect is most useful for fine-tuned networks, which necessarily have reduced diversity due to being trained from the same initial state far from random initialization. Figure 3 (right) and Figure 15 show that when ensembling the original and perturbed networks, ensemble performance indeed scales with Angular CKA dissimilarity. However, fine-tuning ViT models on CIFAR-100 does not share this trend (Figure 21). This contradiction may be explained by observations of similar performance between ensembling and averaging in Utans (1996).

## 4.3. Effect of Hyperparameter Settings

We next compare the stability of different training schemes $\mathcal{T}$ for ResNets trained on CIFAR-10: no weight decay, 10x learning rate warm-up, 4x batch size, Adam, a shallow-wide architecture with similar numbers of parameters (exact details in Table 1).

Figures 4 and 12 show that, in line with prior work (Altıntaş et al., 2023; Vlaar & Frankle, 2022), reducing learning rate (by increasing warm-up) and increasing batch size improve stability. Adam and weight decay reduce stability, which we speculate may be due to their effect on the loss landscape's sharpness, which is known to affect SGD stability (Wu et al., 2018). The shallow-wide architecture is most stable of these settings, which we speculate is due to its training dynamics being more closely aligned with the infinite-width, linearized kernel regime (Lee et al., 2019; Fort et al., 2019).

Next, we consider if a combination of stability-increasing hyperparameters could reduce barriers to 0 for networks

perturbed at initialization. We find that training the shallow-wide architecture combined with 10x learning rate warm-up improves stability over each individual setting (Figure 4 right), but does not eliminate barriers at initialization.

## 4.4. Fine-tuning

Having explored the stability of the loss landscape along trajectories starting from random initialization, we next examine stability on transfer learning trajectories. Fine-tuning is known to have greater stability since it starts from pre-trained networks that have non-random patterns of weights (Neyshabur et al., 2020), but the relative difficulty of the pre-training and transfer task can either increase or decrease stability to training noise (Vlaar & Frankle, 2022).

We again move beyond the effects of training noise to quantify the exact perturbation times and scales at which transfer learning is unstable. We consider task combinations from both vision and language domains, as fine-tuning the latter is known to be unstable to training noise (Juneja et al., 2023).

**Pre-training stability depends on the tasks involved.** Starting with CIFAR-10 and CIFAR-100 (Krizhevsky, 2009), we pre-train two ResNet-50 networks with layer normalization on either task, and then fine-tune them on the opposite task starting from both early (0.24% of pre-training) and late (100% of pre-training) pre-trained checkpoints.

Figure 5 (left, center) and Figures 17 and 18 show that fine-tuning is generally more stable than compared to ResNet-20 (Figure 2) or training the same models from random initialization (Figure 19). This is especially true for later checkpoints ($p = 191ep319st$) and larger perturbations ($\sigma = 0.1$, equivalent to 10% of initialization), whereas fine-tuning from earlier checkpoints is more similar in barriers with regular training (Figure 5). This shows that, as in regular training, instability is mainly a function of pre-training time.

Stability is task-dependent, as transfer from CIFAR-100 to CIFAR-10 is more stable than in reverse. This agrees with Vlaar & Frankle (2022), who find that pre-training on related vs. random data improves or worsens (respectively) the barriers between two points along a training trajectory.[7]

*Vision Transformers (ViTs).* To study a different architecture in the vision domain, we perturb the fine-tuning trajectories of ViTs (Dosovitskiy et al., 2021) of varying sizes on CIFAR-100 (Appendix A.2). While we were only able to consider checkpoints at the end of pre-training, Appendix D.2 shows that, consistent with our previous findings, larger and earlier perturbations during fine-tuning lead to larger barriers.

---

[6]Note that networks with zero barrier still have non-zero Angular CKA, likely because linearly connected networks can perform differently on individual examples (Yunis et al., 2022).

[7]Our work differs in that we consider two diverging trajectories.

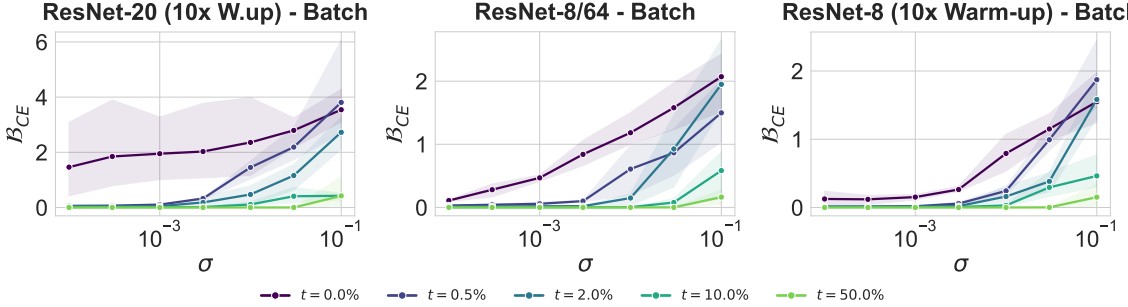

*Figure 4.* Same as Figure 2 (left), but for models trained with 20% warm-up time (**left**), a wider/shallower ResNet8 architecture (**middle**), and both settings (**right**). See Figure 28 for the same plots with log-scaled y-axes, and Appendix C.3 for additional settings.

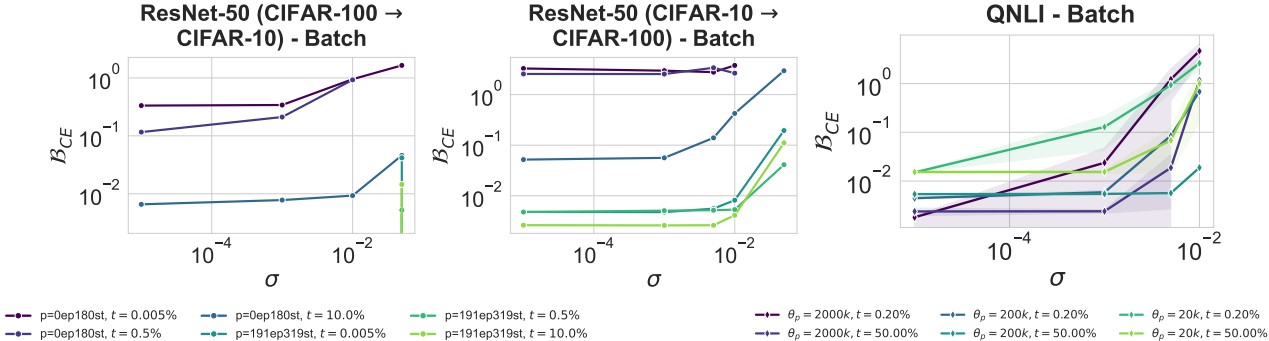

*Figure 5.* Stability of transfer learning on vision tasks: a ResNet-50 is pre-trained on CIFAR-100 and fine-tuned on CIFAR-10 (**left**) or vice versa (**middle**). Barriers (y-axis) are plotted against perturbation magnitudes (x-axis) for various pre-training durations and perturbation times (circle marker colors). See Appendix A.2 for details, and Tables 4-5 for barriers less than $10^{-2}$. **Right**: fine-tuning stability of Multi-BERT on QNLI, starting from 20K, 200K, and 2000K checkpoints with early and late perturbation times (diamond marker colors). For other tasks (MRPC, RTE, SST-2, and CoLA), see Figure 22. For ViT and OLMo, see Figures 20 and 24 respectively.

**Pre-training does not always increase fine-tuning stability.** Having established that longer pre-training improves stability for small-scale vision models, we next examine heavily pre-trained language models. This setting is particularly interesting because recent work by Juneja et al. (2023) demonstrates that, unlike vision models (Neyshabur et al., 2020), training noise can cause language models to converge to distinct basins after fine-tuning.

To investigate this, we analyze the stability of Multi-BERT (Sellam et al., 2022), which provides intermediate checkpoints for every 20,000 steps during pre-training. We take checkpoints at 20k, 200k, and 2000k (100%) steps of pre-training time as starting points and fine-tune on various GLUE tasks (Wang et al., 2019): natural language inference (QNLI, RTE), paraphrase and similarity assessment (MRPC), sentiment classification (SST-2), and linguistic acceptability (CoLA).

Figures 5 and 22 show that BERT is more sensitive to the size of perturbations when compared with our vision experiments (Figure 5 left, middle). For all pre-training checkpoints, earlier perturbations during fine-tuning consistently lead to larger barriers. However, unlike our vision settings, stability does not consistently improve with pre-training time. Notably, for QNLI and RTE (Figure 22), the final pre-trained checkpoint (2000k) has the largest barriers. When evaluating the pre-trained network on these tasks (Table 3), we observe that the 2000k checkpoint has worse test accuracy, despite having lower cross-entropy when compared with the 200k and 20k checkpoints. We speculate that this may be due to overfitting on the pre-training distribution,[8] which could cause the model to become brittle to perturbations during fine-tuning—a phenomenon termed "catastrophic overfitting" by Springer et al. (2025).

*Decoder-only models.* Billion-parameter decoder-only models are widely used in fine-tuning and model merging, but their training dynamics remain severely understudied. To address this gap, we fine-tune intermediate checkpoints of OLMo (Groeneveld et al., 2024) on the math problem dataset GSM8K (Cobbe et al., 2021a). Figure 24 shows that pre-training longer can again reduce stability to fine-tuning, corroborating our MultiBERT findings. Moreover, we observe the same trends—where earlier and larger perturbations result in higher barriers—as in previous settings.

---

[8] MultiBERT was pre-trained for around 100 epochs.

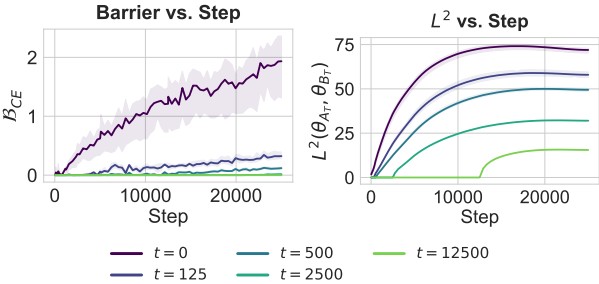

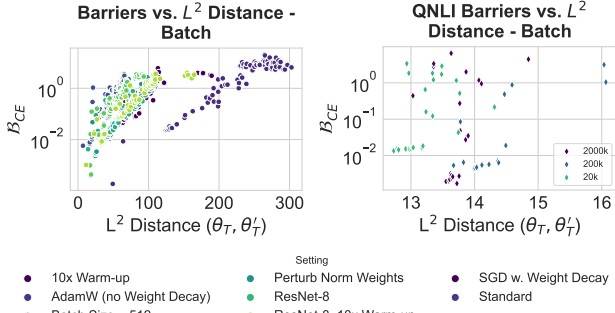

*Figure 6.* Evolution of barriers (**left**) and $L^2$ (**right**) over training for standard ResNet20 trained on CIFAR-10. Each colored line averages over all perturbation magnitudes, as they are nearly indistinguishable.

*Figure 7.* Train loss barriers vs. $L^2$ distance after training between the original and batch-perturbed models for ResNet-20 models trained on CIFAR-10 using various hyperparameter settings (**left**), and BERT models fine-tuned on QNLI (**right**). For additional fine-tuning results, see Figures 21 and 24 to 26 for other BERT tasks, ResNet, ViT, and OLMo respectively.

## 5. $L^2$ Divergence and Barriers

**Barriers and $L^2$ divergence do not evolve according to a linearized dynamical system.** Figure 6 shows the rates at which barriers and $L^2$ divergence increase as training progresses.[9] Contrary to the linearized dynamics derived in Appendix B.4, neither barrier nor $L^2$ increase exponentially over training. More work is needed to explore other mechanisms that could drive these observed rates of divergence.

**Barriers scale with with exponential $L^2$ divergence in vision settings.** Although networks can diverge in weight space without increasing barriers (Frankle et al., 2020a; Vlaar & Frankle, 2022), we find in our experiments that barriers and $L^2$ divergence after training exhibit a strong log-linear relationship (Figure 7 left). This finding differs from Vlaar & Frankle (2022) in that they look at the distance traveled from initialization, whereas we look at the distance between training trajectories which started from the same point. We find that the proportion of identity elements in the aligning permutations $P$ is also related to barriers, albeit to a weaker extent (Figure 16). Since $P$ minimizes $L^2$ distance between $\theta_T$ and $\theta'_T$, this is likely due to the correlation between barriers and $L^2$ divergence.

Interestingly however, fine-tuned language models show little or no correlation between $L^2$ divergence and barriers (Figure 7 right, Figure 26). This suggests that the relationship between $L^2$ and barrier may only appear in smaller-scale models, which highlights the need for large-scale and multi-modal experiments (Juneja et al., 2023).

## 6. Discussion & Conclusion

We present a method for measuring whether neural network training is stable (reliably converging to the same basin),

for a distribution of perturbations, applied at any time in training, on any initial network weights, and for any training procedure. This method allows us to evaluate stability over more conditions, and at a finer precision, than was possible in prior works that only consider the effects of training noise (Vlaar & Frankle, 2022; Fort et al., 2020; Frankle et al., 2020a). Our experiments show that although randomly initialized networks are extremely unstable, stability rapidly increases with training to be robust to perturbations much larger than training noise.

Our work is consistent with the finding in Sarnthein et al. (2023) that a student network initialized very close to a random teacher nevertheless diverges quite far after training. Further work is needed to understand why, unlike in our setting, the student in Sarnthein et al. (2023) remains in the same linearly connected basin as the random teacher.

While instability near initialization is universal, many trends are inconsistent and depend on the task or model. We find that (1) the rate at which stability increases along training trajectories varies greatly, (2) more pre-training does not always improve stability during fine-tuning, (3) $L^2$ divergence correlates strongly with barriers in some cases but not others, and (4) the rates at which $L^2$ and barriers diverge do not match that of a straightforward dynamical system. While the specific counter-examples we have surfaced are sufficient evidence for these results, a detailed exploration of their underlying causes and the circumstances in which they hold (such as in isolating the effects of task versus architecture) is left for future work. Further investigation is also needed to determine (1) if certain hyperparameter settings entirely eliminate instability at initialization, and (2) what perturbations, if any, can be used to reliably improve ensemble performance.

---

[9]Since barriers and $L^2$ are negligible at perturbation time and grow throughout training, this indicates that our results are due to instability and not just the initial perturbation.

## Acknowledgements

Special thanks to Gaurav Iyer, David Mickish, Ekansh Sharma, Sidak Pal Singh, and Julien Boussard for discussions and comments. This research was supported by an NSERC Discovery grant, the Canada CIFAR AI Chairs program, and Fonds de recherche du Québec–Nature et Technologies (FRQNT doctoral research award #352816). Computing resources were provided by Mila–Quebec Artificial Intelligence Institute, and the NVIDIA Corporation.

## Impact Statement

This paper presents work whose goal is to advance the field of Machine Learning. There are many potential societal consequences of our work, none which we feel must be specifically highlighted here.

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

*Table 1.* Hyperparameter settings for ResNet-20 trained on CIFAR-10, along with the test accuracy and training cross-entropy loss of the perturbed model at the end of training. Each setting is averaged over batch and Gaussian perturbations applied at various time steps and scales, with each configuration repeated with three seeds.

| Setting | Model | Optimizer | LR | W-up | WD | BS | Steps | $\text{Acc}_{\text{te}}^1$ | $\text{CE}_{\text{tr}}^1$ |
|---|---|---|---|---|---|---|---|---|---|
| Standard | ResNet20-32 | SGD | 0.100 | 0.020 | – | 128 | 25000 | $0.89 \pm 0.00$ | $0.13 \pm 0.01$ |
| 10x Warm-up | ResNet20-32 | SGD | 0.100 | 0.200 | – | 128 | 25000 | $0.89 \pm 0.01$ | $0.12 \pm 0.01$ |
| AdamW (no Weight Decay) | ResNet20-32 | AdamW | 0.003 | 0.020 | – | 128 | 20000 | $0.89 \pm 0.00$ | $0.13 \pm 0.01$ |
| Batch Size = 512 | ResNet20-32 | SGD | 0.100 | 0.020 | – | 512 | 10000 | $0.88 \pm 0.00$ | $0.15 \pm 0.02$ |
| Perturb Norm Weights | ResNet20-32 | SGD | 0.100 | 0.020 | ✓ | 128 | 20000 | $0.90 \pm 0.00$ | $0.15 \pm 0.00$ |
| ResNet-8 | ResNet8-64 | SGD | 0.100 | 0.020 | – | 128 | 25000 | $0.89 \pm 0.00$ | $0.14 \pm 0.01$ |
| ResNet-8, 10x Warm-up | ResNet8-64 | SGD | 0.100 | 0.200 | – | 128 | 25000 | $0.89 \pm 0.00$ | $0.13 \pm 0.01$ |
| SGD w. Weight Decay | ResNet20-32 | SGD | 0.100 | 0.020 | ✓ | 128 | 20000 | $0.89 \pm 0.01$ | $0.15 \pm 0.01$ |

## A. Training Details

In this section, we provide details about our training methodology. Unless otherwise specified, we conducted all ResNet experiments on individual NVIDIA RTX 8000 GPU with 4 CPU cores. ViT and language model experiments were conducted on NVIDIA L40S GPUs.

### A.1. CIFAR-10 Hyperparameter Experiments

We train residual convolutional models (He et al., 2015) on the CIFAR-10 dataset (Krizhevsky, 2009) using the hyperparameter settings in Table 1. Training times are chosen so that cross-entropy loss on the training data for the perturbed model is below 0.15 after training, on average. All models have test accuracies within 2 percentage points (88-90%). For ease of interpretation, training times are rounded up to the nearest 5000 steps. Although the models in our experiments are not fully converged, and some variations remain between different hyperparameter settings, we did not find our results correlate with different training times, or the network's final train or test performance (Figure 8).

To simplify weight and activation matching, we use layer normalization (Ba et al., 2016) instead of batch normalization, resulting in a slight reduction in performance. When evaluating barriers after permutation alignment, this avoids having to do additional inference passes to correct the batch normalization statistics at each interpolation step (Jordan et al., 2023).

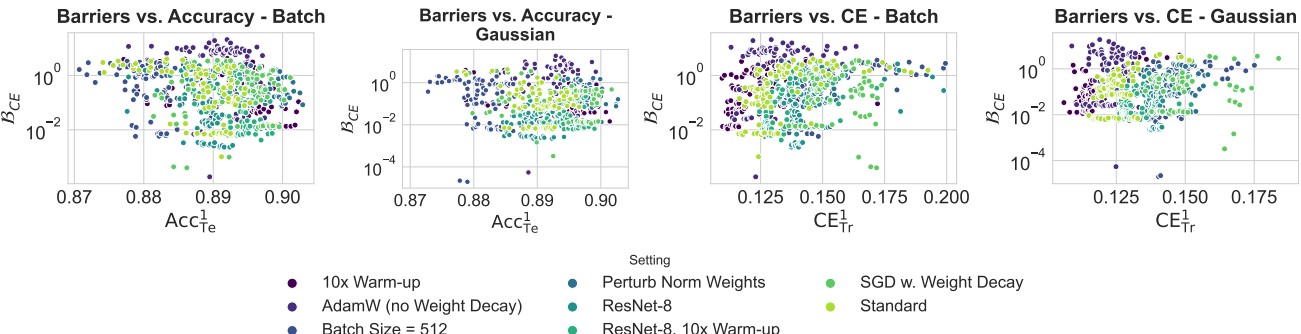

*Figure 8.* Train loss barriers against test accuracy (left) and training cross entropy loss (right) of the perturbed model at the end of training.

### A.2. Finetuning Experiments

**CIFAR pre-training** We pre-trained two ResNet-50 models with different initializations on on both CIFAR-10 and CIFAR-100 datasets, with layer normalization. Each model was trained for 75,000 steps (approximately 200 epochs) using SGD with momentum 0.9 and cosine annealing schedule. We used a peak learning rate of 0.1 with a 2.5% warm-up, with a weight decay of $10^{-4}$ and a batch size of 128. The model was trained with horizontal flips, random rotations up to 10 degrees, random translations up to 4 pixels, and cutout augmentation with 2 pixels.

**CIFAR Fine-tuning**    Starting from our pre-trained ResNet-50 checkpoints, we fine-tuned each model on CIFAR-10 (or CIFAR-100) using stochastic gradient descent with momentum 0.9 for 20,000 steps.

**BERT Fine-tuning Experiments**    We conducted experiments on the GLUE benchmark (Wang et al., 2019) using the MultiBERT model (Sellam et al., 2022), specifically using checkpoints from `google/multiberts-seed_0` and `google/multiberts-seed_1` available on HuggingFace.[10] All tasks share the same base hyperparameters of AdamW (Loshchilov & Hutter, 2017) with a learning rate of $2 \times 10^{-5}$ and a weight decay of 0.01, while the batch size and training duration were scaled according to dataset size, as detailed in Table 2.

Due to computational constraints, we selected QNLI from among the larger datasets with more than 100k examples (QNLI, QQP, MNLI), and followed Devlin et al. (2019) by fine-tuning for three epochs. For the medium-sized SST-2 dataset, we trained for 2,500 steps. Small datasets (RTE, MRPC) with 2.5k–3.7k examples were trained for 500 steps using a batch size of 32, while the medium-small dataset CoLA (5.7k–8.5k examples) was trained for 1,500 steps with the same batch size. These settings ensure that all models achieved a training cross-entropy loss below 0.2, although our networks appear to have overfit the fine-tuning task in some cases (Table 3).

*Table 2.* Task-specific hyperparameters for fine-tuning MultiBERT on GLUE tasks. Training schedule transitions from step-based to epoch-based for larger datasets to ensure sufficient coverage of training data.

| Dataset | Examples | Batch Size | Training Schedule |
|---------|----------|------------|-------------------|
| QNLI | 105k | 128 | 3 epochs |
| SST-2 | 67k | 128 | 2500 steps |
| CoLA | 8.5k | 32 | 1500 steps |
| MRPC | 3.7k | 32 | 500 steps |
| RTE | 2.5k | 32 | 500 steps |

*Table 3.* Multi-BERT (seed 0) test performance of pre-trained checkpoints before fine-tuning (zero-shot evaluation) and after fine-tuning.

| Dataset | Starting Checkpoint | Before Training Acc | CE | End of Fine-tuning Acc | CE |
|---------|---------------------|-----|-----|-----|-----|
| COLA | 2000k | 0.61 | 0.68 | 0.84 | 0.58 |
|      | 200k  | 0.38 | 0.75 | 0.77 | 0.79 |
|      | 20k   | 0.57 | 0.68 | 0.72 | 0.87 |
| MRPC | 2000k | 0.69 | 0.68 | 0.85 | 0.43 |
|      | 200k  | 0.32 | 0.87 | 0.83 | 0.45 |
|      | 20k   | 0.32 | 0.77 | 0.77 | 0.82 |
| QNLI | 2000k | 0.44 | 0.70 | 0.91 | 0.26 |
|      | 200k  | 0.52 | 0.75 | 0.89 | 0.32 |
|      | 20k   | 0.54 | 0.70 | 0.84 | 0.38 |
| RTE  | 2000k | 0.47 | 0.70 | 0.66 | 0.86 |
|      | 200k  | 0.53 | 0.74 | 0.64 | 1.06 |
|      | 20k   | 0.53 | 0.70 | 0.64 | 1.13 |
| SST2 | 2000k | 0.49 | 0.70 | 0.92 | 0.30 |
|      | 200k  | 0.48 | 0.72 | 0.91 | 0.34 |
|      | 20k   | 0.51 | 0.70 | 0.88 | 0.48 |

**ViT Fine-tuning**    We fine-tune on CIFAR-100 starting from four Vision Transformers (ViTs) (Dosovitskiy et al., 2021) on HuggingFace: `google/vit-base-patch16-224` (86M parameters),[11]

---

[10] https://huggingface.co/google/multiberts-seed_0-step_0k and https://huggingface.co/google/multiberts-seed_1-step_0k.

[11] https://huggingface.co/google/vit-base-patch16-224

`google/vit-base-patch16-224-in21k` (86M parameters),[12] `google/vit-large-patch16-224-in21k` (304M parameters),[13] and `google/vit-huge-patch14-224-in21k` (632M parameters).[14] All models were pre-trained on ImageNet-21k, with `vit-base-patch16-224` additionally fine-tuned on ImageNet-1k (Russakovsky et al., 2015).

We use the same hyperparameters across all model sizes: AdamW (Loshchilov & Hutter, 2017) optimizer with learning rate $2 \times 10^{-4}$, weight decay $1 \times 10^{-4}$, batch size 32, and cosine annealing schedule with 10% warm-up over 5 epochs. Data augmentation consisted of horizontal flips, random rotation ($\pm 10°$), random translation ($\pm 16$ pixels), and cutout patches ($16 \times 16$). Images are resized to $224 \times 224$ to match the input resolution expected by the models.

**OLMo Fine-tuning** We fine-tune OLMo-1B[15] on GSM8K (Cobbe et al., 2021b) starting from various checkpoints provided throughout its $\approx 740K$ pre-training steps (3 trillion tokens). For our setting, we select three checkpoints from different training phases: (1) first available checkpoint (4B tokens), (2) mid-way through pretraining (1.5T tokens), and (3) final checkpoint (3T tokens). We fine-tuned each checkpoint for 5,000 steps using AdamW with learning rate of $2 \times 10^{-5}$ and cosine annealing with 10% warm-up.

## B. Methodological Details

In all of our experiments, we train two networks simultaneously with deterministic computations enabled, using identical random seeds for random initialization (if applicable), batch order and data augmentation. We confirm that training with no perturbations results in exactly identical networks as expected.

All evaluations of models trained from initialization are averaged over three runs, while all evaluations for fine-tuned models listed in Appendix A.2 are averaged over two runs.

### B.1. Computing Barriers

To compute barriers, we evaluate 11 equidistant values of $\alpha \in [0, 1]$ along the linear path between $\theta_T$ and $\theta'_T$.

In our definition of barriers (Equation (4)), we interpolate between the loss of the endpoints following (Sharma et al., 2024), rather than taking their average loss as in Frankle et al. (2020a). This is because the former follows from the definition of convexity and is more appropriate for describing a convex loss basin. In practice, since $\theta_T$ and $\theta'_T$ have near-identical loss in our experiments, the definitions are interchangeable.

We also measure test error barriers by replacing $\ell$ with the 0-1 loss over test data. In practice test barriers are slightly less than train barriers as the network reach near-zero loss on the training data but not the test data, allowing for larger barriers in the former. However, since test error barriers follow the same trends as training cross-entropy barriers, we omit them from the text.

### B.2. Computing Angular CKA

There are many different representational similarity methods, and a comparison of them is beyond the scope of this work. We use CKA for a number of reasons: it is invariant to linear transformations (other than affine) which aligns well with the capacity of neural network layers (Kornblith et al., 2019; Lange et al., 2023), it has been used to compare neural networks in other contexts (Nguyen et al., 2021), is less dependent on the weighting of principal components than SVCCA (Raghu et al., 2017), and it can be applied to networks with more intermediate outputs $n_k$ than the number of test inputs $m$ (Kornblith et al., 2019). We report Angular CKA for the simple reason that it gives a distance which increases with dissimilarity, which is in concordance with the other measurements we make. Note that Angular CKA and CKA differ only in the application of arccosine.

---

[12] https://huggingface.co/google/vit-base-patch16-224-in21k
[13] https://huggingface.co/google/vit-large-patch16-224-in21k
[14] https://huggingface.co/google/vit-huge-patch14-224-in21k
[15] https://huggingface.co/allenai/OLMo-1B-hf

We compute the Angular CKA between the final hidden representation as follows:

$$d_{\text{CKA}}(\theta_T, \theta_T')) = \text{CKA}\left[f_{L-1}(\theta_T), f_{L-1}(\theta_T')\right] \tag{5}$$

$$\text{CKA}(\mathbf{X}, \mathbf{Y}) = \arccos\left(\frac{\text{HSIC}(\mathbf{X}, \mathbf{Y})}{\text{HSIC}(\mathbf{X}, \mathbf{X})\,\text{HSIC}(\mathbf{Y}, \mathbf{Y})}\right)$$

where $L$ is the number of residual or attention blocks, $f_{L-1} : \mathbb{R}^{m \times n_0} \to \mathbb{R}^{m \times n_k}$ is the last block's output on a fixed set inputs $\mathcal{X} \in \mathbb{R}^{m \times n_0}$, and HSIC is the Hilbert-Schmidt Independence Criterion, which measures cross correlation between centered similarity matrices.

We use the implementation by Lange et al. (2023), which includes certain modifications to speed computation. Namely, we sample $m = 1000$ examples as Lange et al. (2023) shows CKA can be reliably estimated using reasonably few examples, and we use their reduced-bias estimator for HSIC:

$$\text{HSIC}(\mathbf{X}, \mathbf{Y}) = \frac{2}{m(m-3)}\langle\text{tril}(\mathbf{H}\mathbf{X}\mathbf{X}^\top\mathbf{H}), \text{tril}(\mathbf{H}\mathbf{Y}\mathbf{Y}^\top\mathbf{H})\rangle_F$$

where $m$ is the number of test inputs, $\text{tril}$ extracts the lower triangular portion of a matrix, $\mathbf{H} = \mathbb{I} - \mathbf{1}\mathbf{1}^\top/m$ is a centering matrix that subtracts the mean, and $\langle\cdot, \cdot\rangle_F$ is the Frobenius norm. Effectively, this estimator ignores the diagonal of the similarity matrix $\mathbf{H}\mathbf{X}\mathbf{X}^\top\mathbf{H}$.

### B.3. Perturbation Scale

To ensure fair comparisons between different perturbation methods and network architectures, we normalize all perturbations to have a consistent $L^2$ magnitude, which for ease of interpretation is given relative to the network's size at initialization (Eq. 6). Formally, we ensure the squared norm of $\varepsilon$ matches the total variance at initialization of the perturbed weights, so that

$$\varepsilon = \frac{\hat{\varepsilon} \cdot M}{\|\hat{\varepsilon} \cdot M\|_2}\sqrt{Var[\theta_0 \cdot M]} \tag{6}$$

where $\hat{\varepsilon}$ is a batch or Gaussian perturbation sample, $M$ is a 0-1 mask of the weights to perturb, $\cdot$ is the element-wise product, $Var$ is the expected variance (*not* the sample variance), and $\theta_0$ are the network's initial weights. Thus, for example, a perturbation of magnitude $\sigma = 0.01$ is approximately 1% of the size of $\theta_0$.

Finally, in order to preserve the distribution of activations after each normalization layer, we do not perturb biases or normalization weights in our experiments. While prior work has found that linear mode connectivity can vary between different layers (Vlaar & Frankle, 2022; Zhou et al., 2023; Adilova et al., 2024), we did not find that our results changed significantly depending on which layers were perturbed. Figure 9 shows results when only perturbing biases and normalization weights. In this case, we use the same scale of perturbations as we would normally assign to the weights in the layer following the biases or normalization weights.

### B.4. Linearized Approximation For $L^2$ Divergence

A classical result of dynamical systems states that a linearized system subject to a small perturbation can diverge exponentially with respect to time at a rate which depends on the largest eigenvalue (i.e. top Lyapunov exponent) of the gradient of the training map (Strogatz, 2019):

$$\mathcal{T}(\theta_i + \varepsilon) \approx \mathcal{T}(\theta_i) + \varepsilon^\top\nabla_\mathcal{T}\theta_i,$$

$$\mathcal{T}^T(\theta_0 + \varepsilon) \approx \mathcal{T}^T(\theta_0) + \varepsilon^\top\prod_{t=1}^{T}\nabla_\mathcal{T}\theta_0$$

$$\|\theta_T - \theta_T'\|_2 \leq \|\varepsilon\|_2 e^{\lambda t} \tag{7}$$

where $\nabla_\mathcal{T}\theta_0$ is the gradient of the training map with respect to the weights and $\lambda$ is the top eigenvalue over all the gradients at each step $t$.

Substituting in the definition of SGD, we find that the $L^2$ divergence between the original and perturbed models after

training depends on the curvature of the loss landscape:

$$\mathcal{T}(\theta_i) = \theta_i - \eta \nabla_\ell \theta_i, \qquad \nabla_\mathcal{T} \theta_i = I - \eta_i H_i,$$
$$\|\theta_T - \theta'_T\|_2 \leq \|\varepsilon\|_2 e^{\lambda_H t} \tag{8}$$

where $\eta_i$ is the learning rate, $H_i$ is the Hessian of the weights, and $\lambda_H$ is the largest eigenvalue over all $I - \eta_i H_i$. Divergence results either from high positive curvature as in Wu et al. (2018), but additionally if there is any negative curvature in the perturbation direction, and $|\lambda_H| > 1$ implies the possibility of exponential growth in divergence over training.

## C. Further Experiments

### C.1. Baselines

*Stability is not specific to layer type.* In the main text (Figure 2), we excluded norm weights and biases from perturbations. However, in Figure 9, we show that perturbing only norm layers leads to similar trends. This suggests that fine-tuning stability is influenced more by overall network dynamics rather than specific layer types. While individual parameters or layers may have varying importance as noted in previous work (Adilova et al., 2024), batch perturbations already capture this effect to some extent. The increased barriers for the smallest perturbations at initialization are an artifact of numerical instability—in our regular experiments, we avoid this problem by reducing the fraction of perturbed weights instead of reducing the perturbation scale beyond $\sigma = 10^{-4}$.

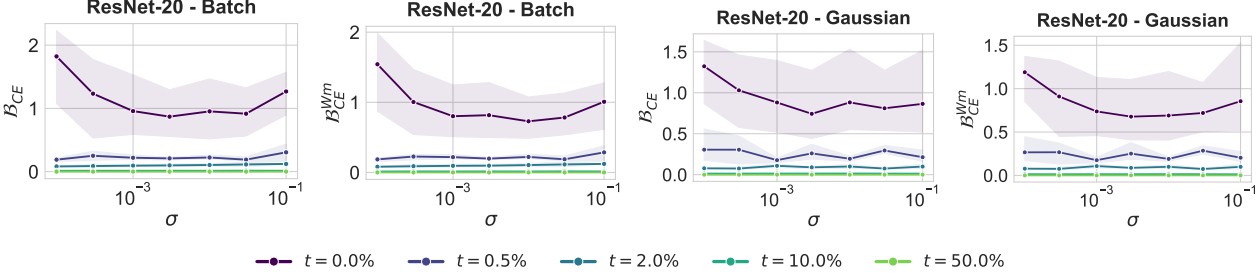

*Figure 9.* Train loss barriers before and after permutations when perturbing only normalization layers. Results are shown for batch (left) and Gaussian (right) perturbations on ResNet-20 trained with SGD (momentum, no weight decay), using a learning rate of 0.1, 2% warm-up, and a batch size of 128 for 20,000 steps.

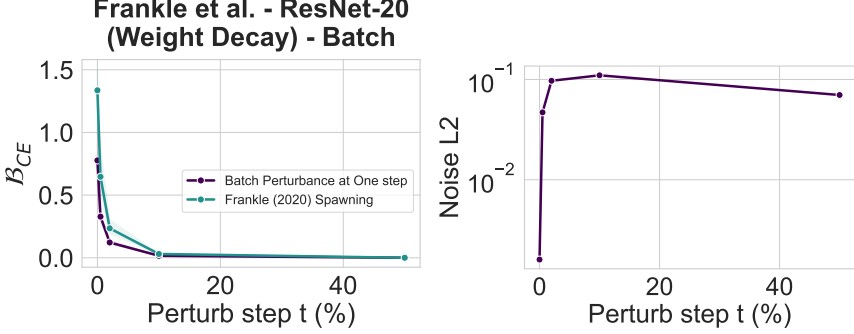

*Figure 10.* **Left:** Comparison of training loss barriers between our butterfly setting and Frankle et al.'s spawning setting. In the spawning setting, each network is trained with different non-determinism after perturbation step $t$, while our method applies a single perturbation. As expected, our single-perturbation approach provides a lower bound on the spawning barriers. **Right:** $L^2$ magnitude of the expected deviation between two copies of the same model, when each model takes a single independent training step at time $t$.

*How do our perturbations compare to SGD noise?* To establish the relative magnitude of our perturbations vs. SGD noise, we replicate the parent-child spawning experiment of Frankle et al.. Figure 10 shows that our batch perturbations are a lower

bound on the Frankle baseline's barriers, meaning that the instability resulting from training independently for multiple steps must be at least the instability resulting from a single independent training step.

Note that for this comparison, we scale batch perturbations to the expected magnitude of SGD noise at the perturbation time $t$. This makes batch perturbation equivalent to taking only one step at time $t$ with different SGD noise, as opposed to using different SGD noise from $t$ onwards in the Frankle baseline.

### C.2. Perturbing Only A Fraction of Weights.

We further decrease the scale of our perturbations from Figure 2 by perturbing only a fraction of the weights with our smallest perturbation scale of $10^{-4}$. Strikingly, we find that perturbing as little as a **single weight**, which occurs when the fraction of perturbed weights is $10^{-6}$, is sufficient to create barriers at initialization (Figure 11, right). The scale of this perturbation (Figure 11, left) is well below that of noise caused by hardware indeterminacy.

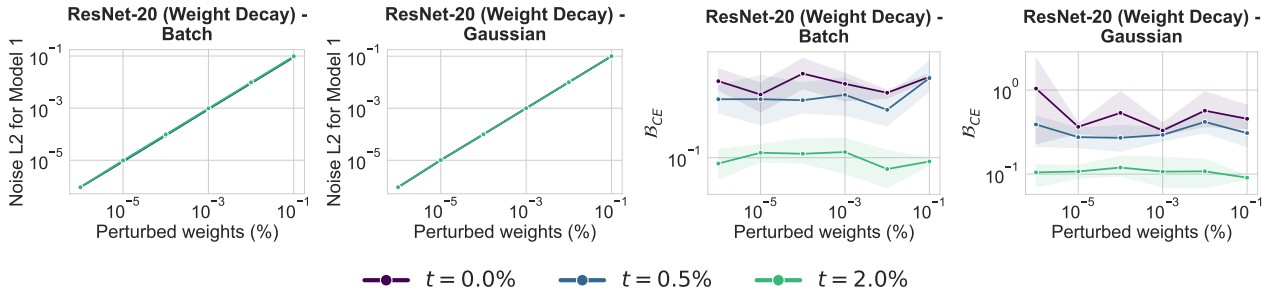

Figure 11. **Left:** absolute $L^2$ norm of the noise as a function of the fraction of perturbed weights. **Right:** train loss barriers as a function of the fraction of perturbed weights.

### C.3. Additional Hyperparameter Settings

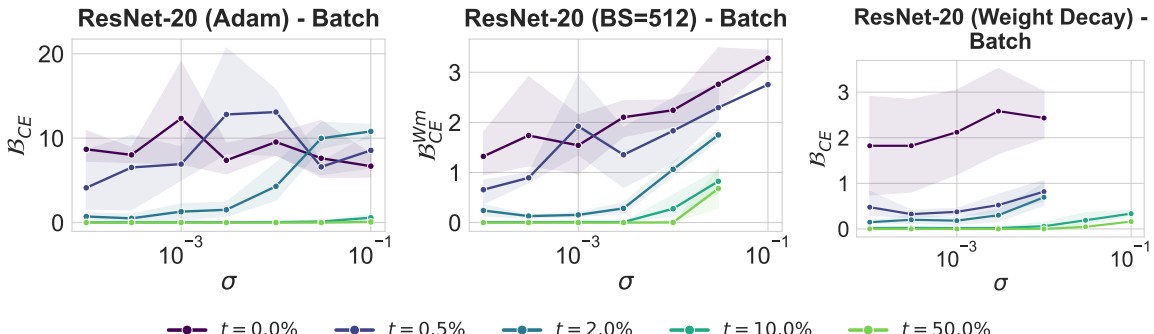

Figure 12. Same as Figure 2 with AdamW without weight decay and learning rate of 0.003 (**left**), batch size of 512 (**middle**), and weight decay (**right**).

In this section we present the results of training ResNet-20 on CIFAR-10 for all hyperparameter combinations listed in Table 1.

It is well-known that neural network training is highly sensitive to optimization hyperparameters (Smith, 2018). Our experiments corroborate the complex interdependencies between key optimization hyperparameters, suggesting that further theoretical exploration is needed. We specifically focus on the impact of optimizer choice, batch size, and weight decay.

*SGD enhances training stability.* The choice of optimizer significantly impacts the stability of the training map. Additionally, networks trained with Adam also exhibit higher $L^2$ distances (Figure 7). We stipulate that this phenomenon is linked to the

implicit bias of SGD. Specifically, Bradley et al. (2022) highlight that SGD's inherent noise helps it avoid high-curvature regions, which are often linked to poor generalization and, in our setting, could likely contain linearly connected minima.

In our experiments, we observed that larger batch sizes lead to higher loss barriers, suggesting a trade-off between batch size and stability. This phenomenon aligns with findings from Keskar et al. (2017), who show that larger batch sizes can lead to sharper minima, which might increase the loss barriers and potentially hurt generalization.

Weight decay provides slightly more stability, particularly right after initialization. D'Angelo et al. (2024) highlight that weight decay modifies the optimization dynamics, enhancing the implicit regularization of SGD through loss stabilization mechanisms. They also emphasize the role of gradient clipping, which stabilizes training, particularly at the edge of stability.

### C.4. Functional Diversity For Additional Hyperparameter Settings

We replicate Section 4.2 on two of our additional hyperparameter settings: with weight decay, and with 10x warm-up. The results in Figures 13 to 15 closely agree with our findings in Figure 3.

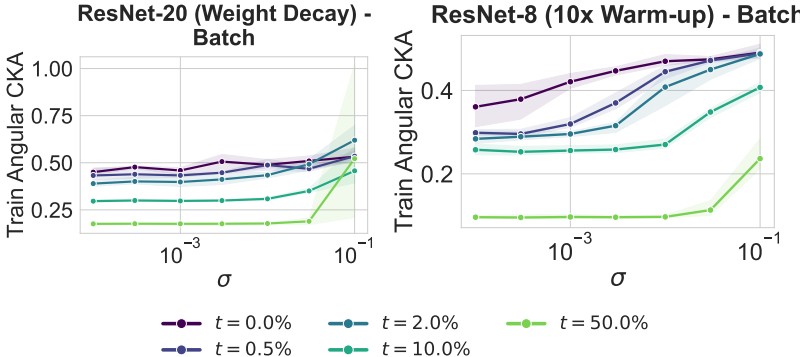

*Figure 13.* Representational similarity distance measured via Angular CKA for ResNet-20 with weight decay (**left**) or 10x warm-up (**right**).

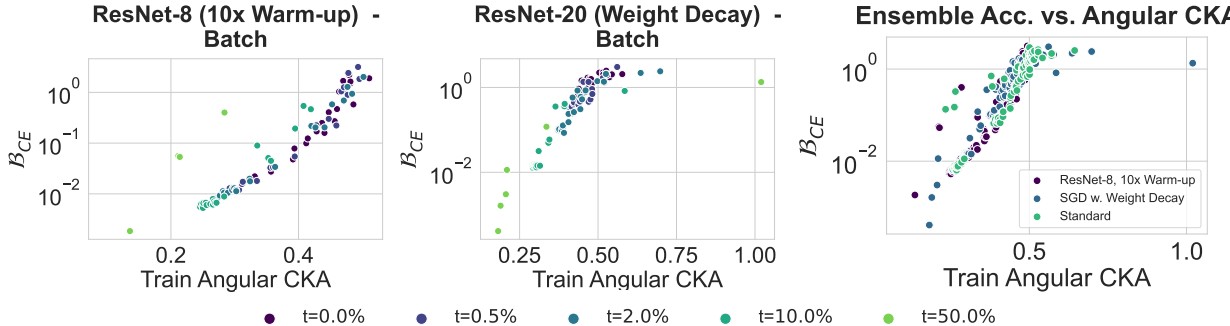

*Figure 14.* Train loss barrier versus representational similarity distance on ResNet-20 with weight decay (**left**), ResNet-8 with 10x warm-up (**middle**) with perturbation time indicated by color, and including the standard setting from Figure 3, with settings indicated by color (**right**).

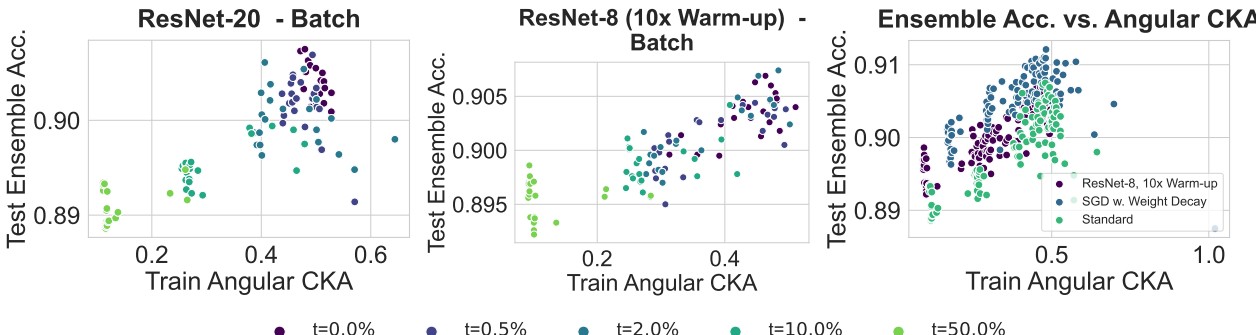

*Figure 15.* Test accuracy of an ensemble of the original and perturbed models after training, versus representational similarity distance, on ResNet-20 with weight decay (**left**), ResNet-8 with 10x warm-up (**middle**) with perturbation time indicated by color, nd including the standard setting from Figure 3, with settings indicated by color (**right**).

### C.5. Fixed Points of Aligning Permutations

While weight matching is unable to reduce barriers in our case of *identically* initialized networks, we investigate whether the underlying mechanism proposed by (Entezari et al., 2022)—that barriers arise from network permutations—is still relevant to our observations. We consider if the observed barriers and $L^2$ distances between the original and perturbed networks correlate with the number of fixed points in the permutations found by weight alignment (Ainsworth et al., 2023). Here, fixed points refer to the un-permuted elements in the aligning permutations. Since we expect two networks with identical weights to be aligned by permutations consisting only of fixed points, the fraction of fixed points can be used to indicate the degree to which two diverging networks have been permuted with respect to one another.

Figure 16 suggests a weak correlation, where the number of fixed points is inversely proportional to both the barrier heights and the $L^2$ distance between the networks before alignment in the ResNet settings. Unfortunately, this observation does not extend to our ViT or BERT settings, as weight matching fails to identify non-trivial permutations (i.e., permutations other than the identity) for these transformer architectures.

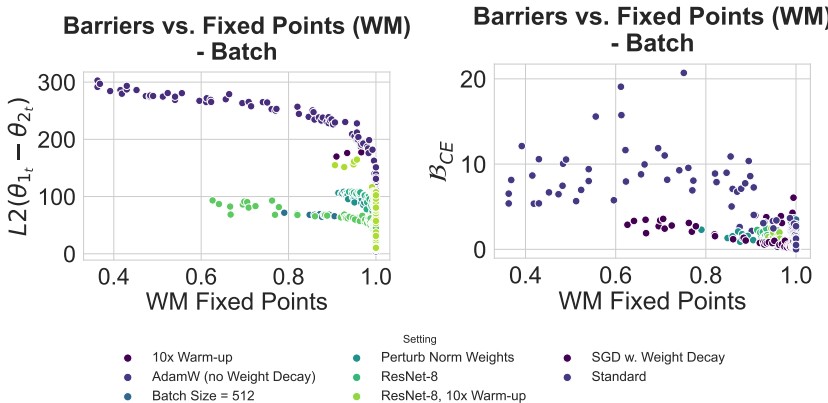

*Figure 16.* Relationship between the fraction of fixed points (un-permuted elements) in the weight matching permutation aligning the ResNet-20 models trained on CIFAR-10 from Figure 2, and $L^2$ divergence (**left**) or train barriers (**right**).

## D. Further Fine-tuning Results

### D.1. ResNet Fine-Tuning

*Extended pre-training yields near-zero fine-tuning barriers in ResNet-50/CIFAR settings.* Figure 5 shows that fine-tuning from later checkpoints greatly improves stability in ResNet-50 experiments. For improved readability, we zoom into the perturbations with $\sigma \leq 10^{-2}$ in Figure 18 and Tables 4 and 5, revealing that transferring from CIFAR-100 to CIFAR-10

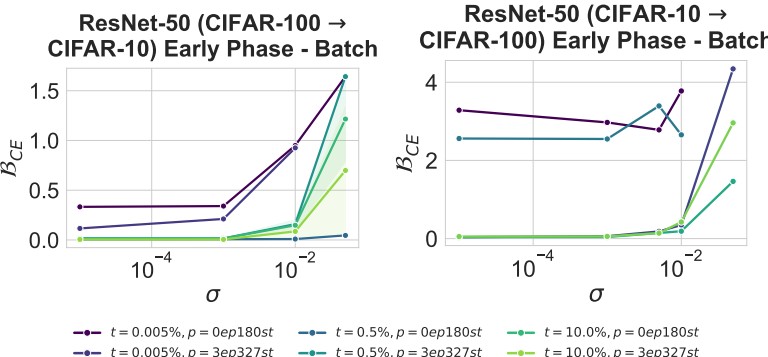

*Figure 17.* Same as Figure 5 (left), but with additional pre-training times and perturbation times from early in training. Stability of transfer learning on vision tasks: a ResNet-50 is pre-trained and fine-tuned (see Appendix A.2 for details) from CIFAR-100 to CIFAR-10 (**left**) or vice versa (**right**). Barriers (y-axis) are plotted against perturbation magnitudes (x-axis) for various combinations of initial pre-trained weights and perturbation times (colors).

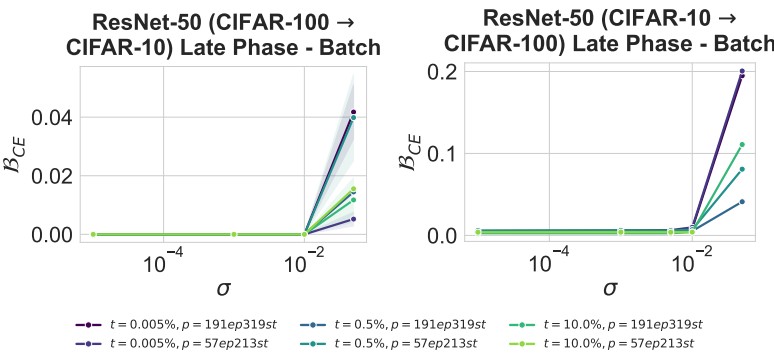

*Figure 18.* Same as Figure 5 (left) and Figure 17, but with additional pre-training times and perturbation times from late in training. Barriers at scales $< 10^{-2}$ are near-zero (see Tables 4 and 5 for exact values).

and vice-versa results in near-zero barriers. The CIFAR-100 to CIFAR-10 direction exhibits greater stability, which could indicate that CIFAR-100 pre-training is better suited for optimization on CIFAR-10 than the other way around.

*Random initializations are less stable than pre-trained initializations.* In Figure 18, we demonstrated that pre-training improves fine-tuning stability, while Figure 17 suggested that earlier checkpoints are more brittle to perturbations. As a baseline, we train a ResNet-50 from random initialization on CIFAR-10 and find that it exhibits even larger barriers (Figure 19), with a similar magnitude to training from scratch.

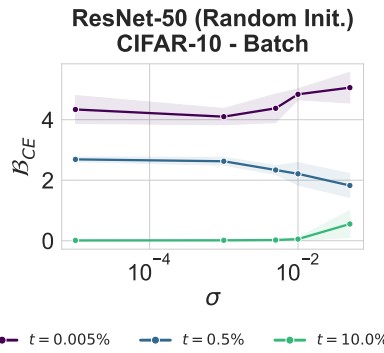

*Figure 19.* Same as Figures 5, 17 and 18 (left panel of each), but with randomly initialized ResNet-50 instead of networks pre-trained on CIFAR-100. Train loss barriers are reported for batch perturbations during fine-tuning on CIFAR-10 using the recipe from Appendix A.2.

Table 4. Train loss barriers and $L^2$ distance in Cifar-100 $\rightarrow$ Cifar-10 setting. Same as Figure 18 (left).

| Starting Checkpoint | Relative Perturb Step (%) | $\sigma$ | $L^2$ Distance | Train $\mathcal{B}_{\text{CE}}$ |
|---|---|---|---|---|
| 191ep319st | 0.005 | 0.00001 | 39.41 ± 0.23 | 0.00 ± 0.00 |
| 191ep319st | 0.005 | 0.00100 | 39.46 ± 0.18 | 0.00 ± 0.00 |
| 191ep319st | 0.005 | 0.01000 | 40.41 ± 0.82 | 0.00 ± 0.00 |
| 191ep319st | 0.5 | 0.00001 | 39.49 ± 0.23 | 0.00 ± 0.00 |
| 191ep319st | 0.5 | 0.00100 | 39.42 ± 0.19 | 0.00 ± 0.00 |
| 191ep319st | 0.5 | 0.01000 | 40.03 ± 0.62 | 0.00 ± 0.00 |
| 191ep319st | 10 | 0.00001 | 38.02 ± 0.18 | 0.00 ± 0.00 |
| 191ep319st | 10 | 0.00100 | 38.04 ± 0.21 | 0.00 ± 0.00 |
| 191ep319st | 10 | 0.01000 | 38.19 ± 0.13 | 0.00 ± 0.00 |
| 57ep213st | 0.005 | 0.00001 | 40.81 ± 0.14 | 0.00 ± 0.00 |
| 57ep213st | 0.005 | 0.00100 | 40.93 ± 0.15 | 0.00 ± 0.00 |
| 57ep213st | 0.005 | 0.01000 | 41.49 ± 0.44 | 0.00 ± 0.00 |
| 57ep213st | 0.5 | 0.00001 | 40.90 ± 0.06 | 0.00 ± 0.00 |
| 57ep213st | 0.5 | 0.00100 | 40.82 ± 0.17 | 0.00 ± 0.00 |
| 57ep213st | 0.5 | 0.01000 | 41.26 ± 0.29 | 0.00 ± 0.00 |
| 57ep213st | 10 | 0.00001 | 39.50 ± 0.14 | 0.00 ± 0.00 |
| 57ep213st | 10 | 0.00100 | 39.50 ± 0.13 | 0.00 ± 0.00 |
| 57ep213st | 10 | 0.01000 | 39.63 ± 0.10 | 0.00 ± 0.00 |

Table 5. Train loss barriers and $L^2$ distance in Cifar-10 $\rightarrow$ Cifar-100 setting. Same as Figure 18 (right).

| Starting Checkpoint | Relative Perturb Step (%) | $\sigma$ | $L^2$ Distance | Train $\mathcal{B}_{\text{CE}}$ |
|---|---|---|---|---|
| 191ep319st | 0.005 | 0.00001 | 67.98 ± 0.00 | 0.00 ± 0.00 |
| 191ep319st | 0.005 | 0.00100 | 68.05 ± 0.00 | 0.00 ± 0.00 |
| 191ep319st | 0.005 | 0.00500 | 68.92 ± 0.00 | 0.01 ± 0.00 |
| 191ep319st | 0.005 | 0.01000 | 70.73 ± 2.25 | 0.01 ± 0.00 |
| 191ep319st | 0.5 | 0.00001 | 67.89 ± 0.00 | 0.00 ± 0.00 |
| 191ep319st | 0.5 | 0.00100 | 67.96 ± 0.00 | 0.01 ± 0.00 |
| 191ep319st | 0.5 | 0.00500 | 68.18 ± 0.00 | 0.01 ± 0.00 |
| 191ep319st | 0.5 | 0.01000 | 68.34 ± 0.25 | 0.01 ± 0.00 |
| 191ep319st | 10 | 0.00001 | 64.05 ± 0.00 | 0.00 ± 0.00 |
| 191ep319st | 10 | 0.00100 | 64.11 ± 0.00 | 0.00 ± 0.00 |
| 191ep319st | 10 | 0.00500 | 64.47 ± 0.00 | 0.00 ± 0.00 |
| 191ep319st | 10 | 0.01000 | 65.54 ± 0.94 | 0.00 ± 0.00 |
| 57ep213st | 0.005 | 0.00001 | 62.34 ± 0.00 | 0.01 ± 0.00 |
| 57ep213st | 0.005 | 0.00100 | 62.68 ± 0.00 | 0.01 ± 0.00 |
| 57ep213st | 0.005 | 0.00500 | 63.31 ± 0.00 | 0.01 ± 0.00 |
| 57ep213st | 0.005 | 0.01000 | 65.23 ± 2.39 | 0.01 ± 0.00 |
| 57ep213st | 0.5 | 0.00001 | 62.32 ± 0.00 | 0.01 ± 0.00 |
| 57ep213st | 0.5 | 0.00100 | 62.34 ± 0.00 | 0.01 ± 0.00 |
| 57ep213st | 0.5 | 0.00500 | 62.70 ± 0.00 | 0.01 ± 0.00 |
| 57ep213st | 0.5 | 0.01000 | 63.21 ± 0.72 | 0.01 ± 0.00 |
| 57ep213st | 10 | 0.00001 | 59.47 ± 0.00 | 0.00 ± 0.00 |
| 57ep213st | 10 | 0.00100 | 59.43 ± 0.00 | 0.00 ± 0.00 |
| 57ep213st | 10 | 0.00500 | 59.50 ± 0.00 | 0.00 ± 0.00 |
| 57ep213st | 10 | 0.01000 | 59.70 ± 0.29 | 0.00 ± 0.00 |

## D.2. ViT Fine-Tuning

Initial fine-tuning stability experiments on ResNets and MultiBERTs suggested opposed trends between vision tasks with convolutional architectures, and language tasks with tarnsformer architectures. To disentangle whether this difference is due to the nature of the task or architecture, we applied our experimental procedure to Vision Transformers (ViTs), which are representative of the transformer architecture and larger-scale models than ResNet-50. The sources for the pre-trained ViTs and the fine-tuning procedure we use are specified in Appendix A.2.

While we did not have access to intermediate training checkpoints for ViT models, we instead compare four ViTs of different sizes which were pre-trained on ImageNet variants: google/vit-base-patch16-224 (86M parameters), google/vit-base-patch16-224-in21k (86M parameters), google/vit-large-patch16-224-in21k (304M parameters), and google/vit-huge-patch14-224-in21k (632M parameters). The size of each pre-training dataset serves as a proxy for pre-training duration.

*Extending our findings to ViTs fine-tuned on CIFAR-100.* Consistent with our findings on smaller convolutional networks, in Figure 20 we observe that earlier and larger perturbations cause more pronounced barriers. Interestingly, the google/vit-base-patch16-224 variant, which underwent additional fine-tuning on ImageNet-1K after its initial ImageNet-21K pre-training, effectively represents a longer training process. Although the exact learning rate schedule is unknown, this setting resembles the extended pre-training observed in later BERT checkpoints. Models fine-tuned from this variant (Figure 20, top left) exhibit the largest barriers among all settings, with barriers one order of magnitude larger than those from google/vit-base-patch16-224-in21k (Figure 20, top right). This provides additional evidence that extended pre-training reduces fine-tuning stability. Future work could investigate the stability of intermediate ViT checkpoints.

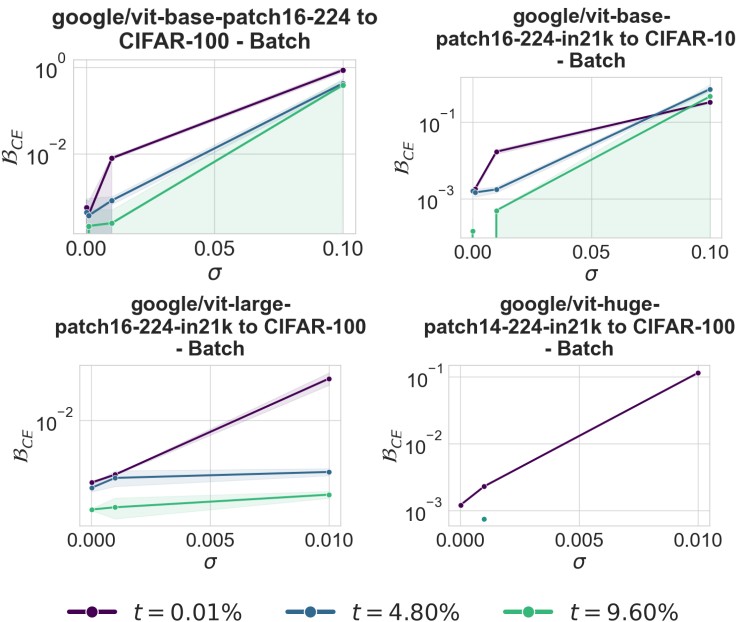

*Figure 20.* Stability of various ViT architectures on CIFAR-100. Training loss barriers after training (y-axis) are plotted against perturbation magnitude (x-axis) and perturbation step (color). We fine-tune four pre-trained ViTs of different sizes: ViT-Base/16-224 (86M parameters, pre-trained on ImageNet-21K and then fine-tuned on ImageNet), ViT-Base/16-224-in21k (86M parameters pre-trained on ImageNet-21K), ViT-Large/16-224-in21k (304M parameters), and ViT-Huge/14-224-in21k (632M parameters).

## D.3. BERT Fine-Tuning

Figure 22 plots additional fine-tuning tasks as described in Appendix A.2.

*Representational similarity for BERT models.* We also provide Angular CKA plots for BERT on MRPC and QNLI datasets in Figure 23. Figure 23 shows that barriers are correlated with Angular CKA, indicating real functional differences between the networks. This is consistent with our findings in vision models (Figures 3, 14 and 21 right), unlike the correlation

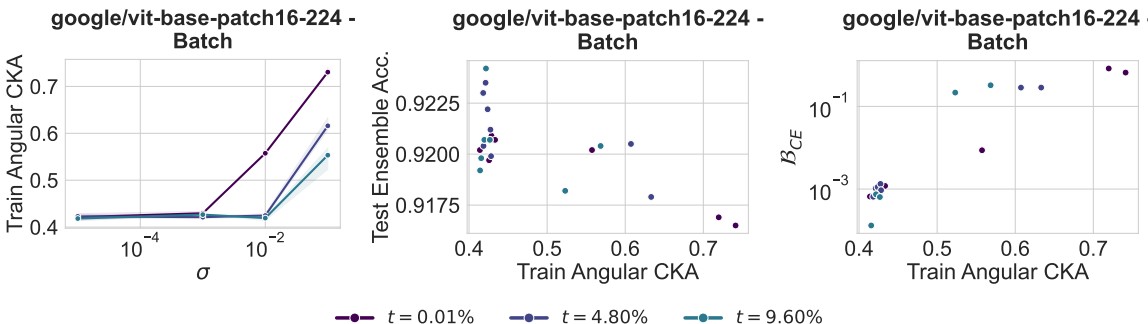

*Figure 21.* **Left:** representational similarity distance measured via Angular CKA for ViT-base. **Middle:** test ensembling accuracy against Angular CKA. **Right:** training loss barrier against Angular CKA.

between barriers and $L^2$ divergence (Figure 26) which is not consistent between vision and language settings.

### D.4. OLMo Fine-Tuning

Figure 24 demonstrates that the trends we observe in terms of perturbation time and magnitude extend to decoder-only large language models. Consistent with our MultiBERT findings, we find that more pre-training does not necessarily lead to improved fine-tuning stability.

### D.5. $L^2$ Divergence and Barriers

Here, we examine the relationship between the barriers and the $L^2$ divergence between models at the end of training in greater detail. Figure 25 (left, middle) shows that fine-tuning on vision tasks, such as transferring from CIFAR-100 to CIFAR-10 and vice versa, follows the trends presented in Figure 7 (left). We see that this direct relationship is weak or non-existent when transferring ViTs from ImageNet to CIFAR-100 (Figure 25 right), as well as for the GLUE benchmark in our study (Figure 26), with QNLI and COLA exhibiting almost no correlation. Interestingly, OLMo fine-tuned on GSM8K (Figure 24, right) shows a clearer correlation between barriers and $L^2$ divergence, which more closely resembles our ResNet-20 results (Figure 7, left) than BERT.

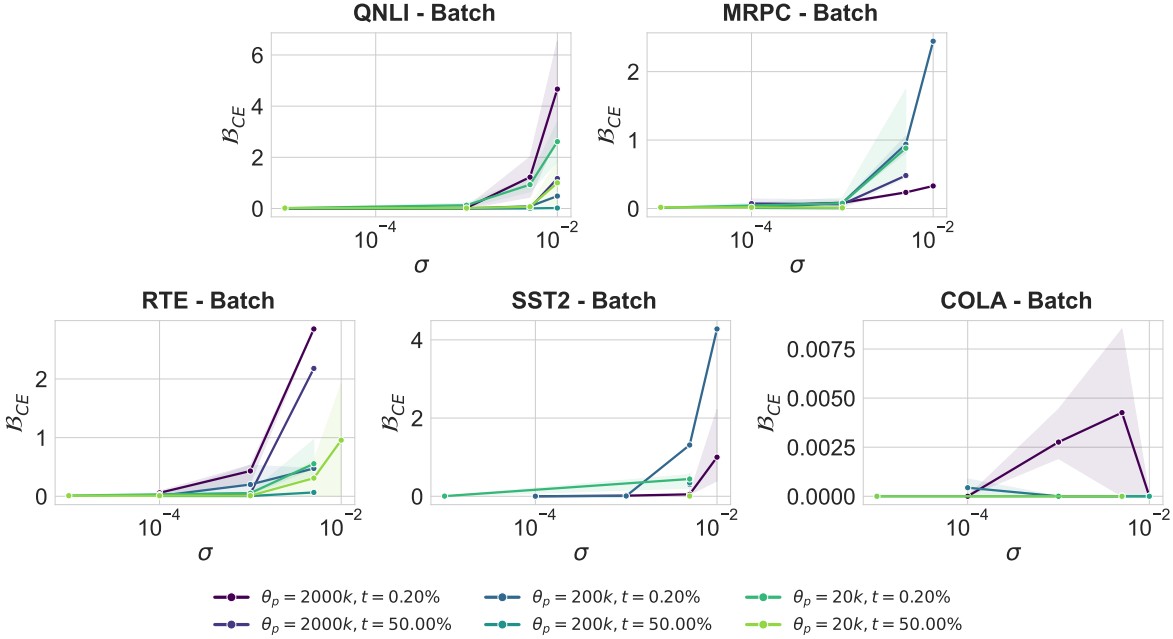

*Figure 22.* Same as Figure 5 (right), but with additional pre-training times, perturbation times, and tasks. Fine-tuning stability of Multi-BERT on QNLI and MRPC, starting from 20K, 200K, and 2000K checkpoints with different perturbation times. Tasks are QNLI (**top left**), MRPC (**top right**), RTE (**bottom left**), SST-2 (**bottom middle**), and COLA (**bottom right**).

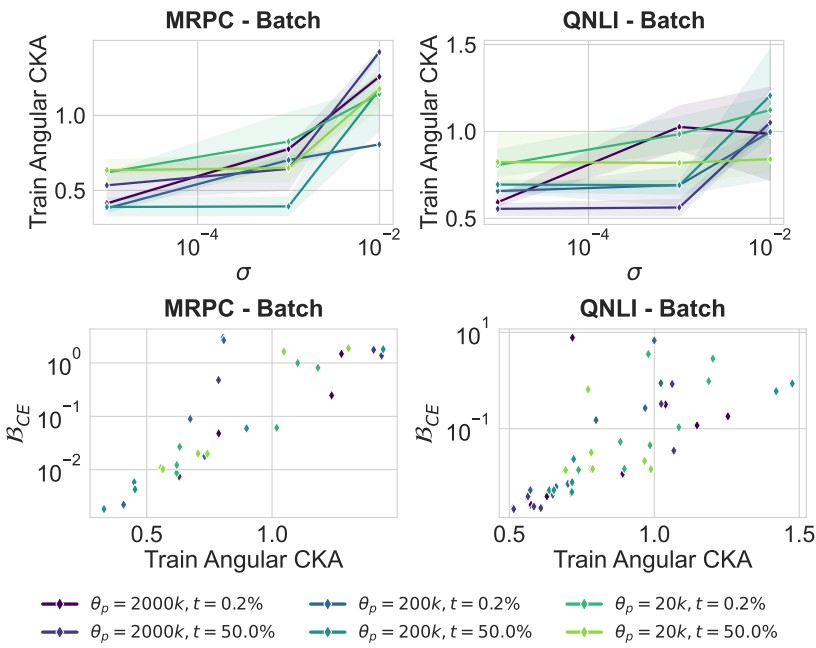

*Figure 23.* **Top:** representational similarity distance measured via Angular CKA for MultiBERT on MRPC (**left**) and QNLI (**right**). **Bottom:** barriers vs. angular CKA on MRPC (**left**) and QNLI (**right**).

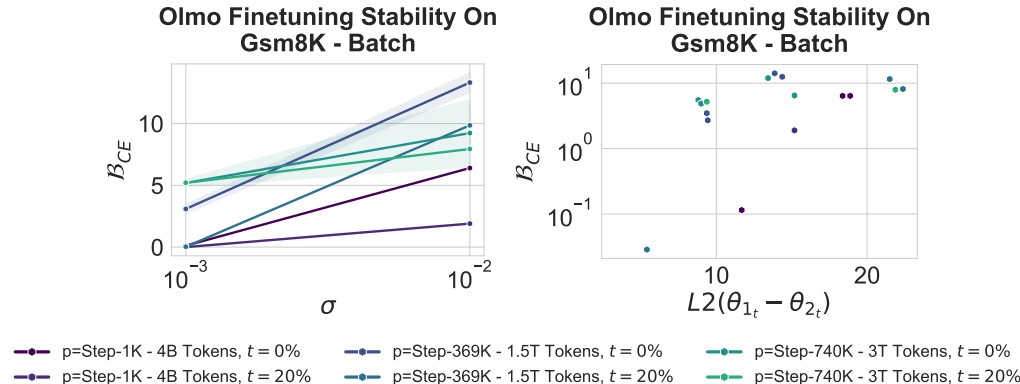

*Figure 24.* Stability of fine-tuning OLMo-1B on GSM8K mathematical reasoning tasks. We fine-tune OLMo-1B checkpoints from different pre-training stages (early, middle, and final checkpoints) on GSM8K with batch perturbations applied at various training steps. **Left:** Loss barriers (y-axis) plotted against perturbation magnitude (x-axis) for different checkpoint combinations and perturbation steps (colors). **Right:** Barriers vs. $L^2$ distance between the original and perturbed models. Consistent with our vision experiments, earlier perturbations and later pre-training checkpoints lead to higher loss barriers, demonstrating that fine-tuning stability patterns generalize from vision to language models.

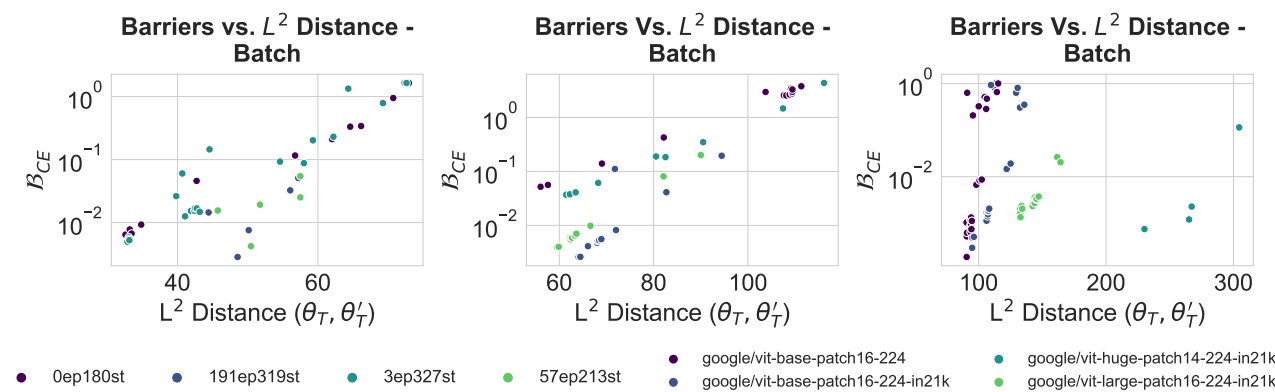

*Figure 25.* Train loss barriers vs. $L^2$ distance between the perturbed and original models at the end of training for fine-tuning vision models: ResNet-50 transferring from CIFAR-100 to CIFAR-10 (**left**), ResNet-50 transferring from CIFAR-10 to CIFAR-100 (**middle**), and ViT-base fine-tuned on CIFAR-100 (**right**). Note the legend for the colors is different in the rightmost plot.

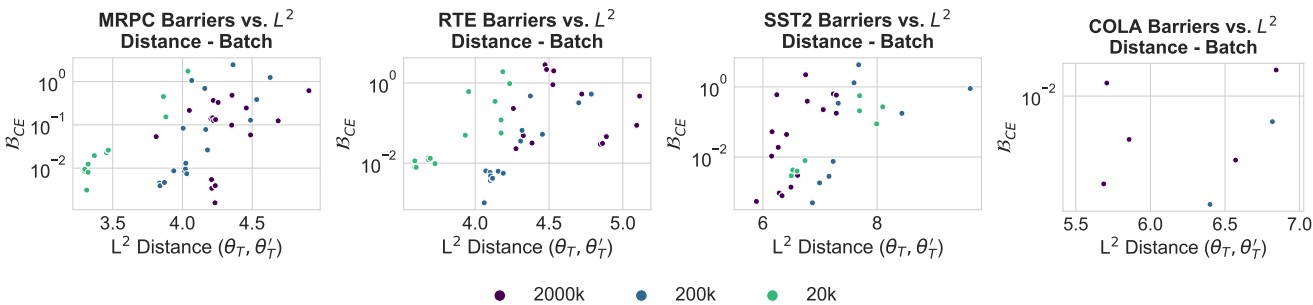

*Figure 26.* Train loss barriers vs. $L^2$ distance between the perturbed and original models at the end of training for MRPC, RTE, SST-2, and COLA.

# E. Additional Log-Scale Plots

These plots are copies of main figure plots with log y-axes, and are included to display a clearer separation between smaller barriers.

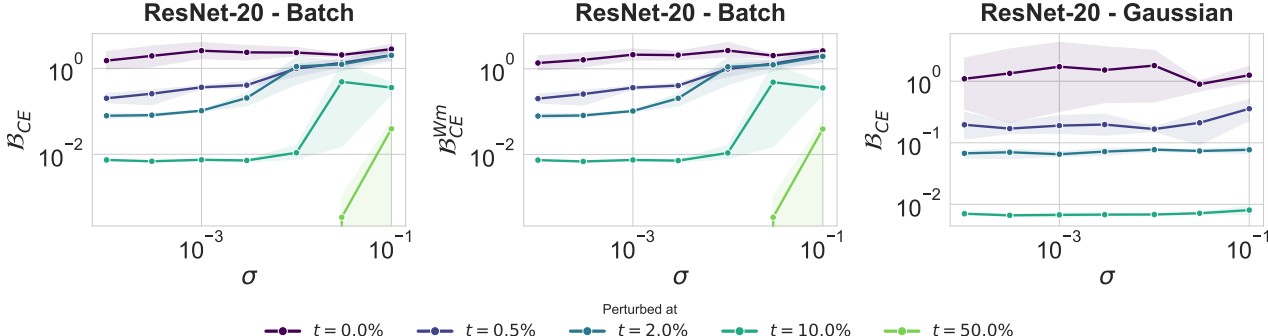

*Figure 27.* Same data as Figure 2 with y-axis in log-scale to improve the readability of smaller barriers. Loss barriers on training data at the end of training (y-axis) are plotted against perturbation magnitude (x-axis) and perturbation step (color indicates fraction of total training time). **Left:** barriers due to batch perturbation. **Middle:** batch perturbation barriers after accounting for permutations. **Right:** barriers due to Gaussian perturbation.

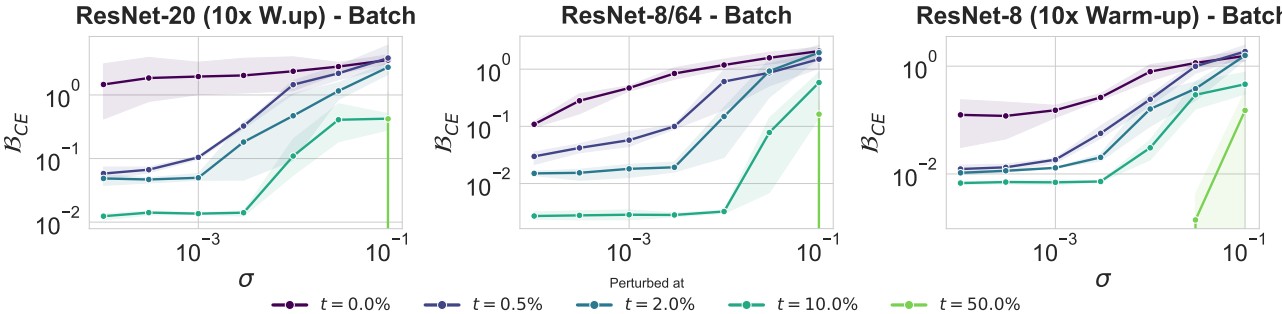

*Figure 28.* Same as Figure 4 but with y-axis in log-scale for models trained with 20% warm-up time (**left**), a wider/shallower ResNet8 architecture (**middle**), and a combination of both settings (**right**).

