# OpenReview forum: "The Butterfly Effect: Neural Network Training Trajectories Are Highly Sensitive to Initial Conditions"
_ICML.cc/2025/Conference — ICML 2025 poster_

### Official Review · Reviewer_mmJd · 2025-03-09

**Overall Recommendation:** 3

**Summary:**

The submission investigates the sensitivity of neural network training outcome to initial conditions. Building upon related work in optimization and training dynamics, it investigates the conditions for stability and identifies a chaotic early stage in which small perturbations cause trajectory divergence. Experimental evaluations demonstrate that effect in vision models, but find different effects in language models.

### Summary after Rebuttal
In the rebuttal, the authors have added behavioral comparisons and removed claims with weak support, which I believe improve the submission overall. I encourage the authors to continue improving the experimental support and will keep my score as is for now.

**Claims And Evidence:**

Before I go into this, I generally like the paper (see strenghts and weaknesses below). However, I find the claims made way to general for the experimental support.

Experiments are performed mostly on relatively small vision models with relatively simple tasks. I understand the need for tractable experiment setups, but the experimental evaluation would be strengthened significantly if showed beyond ResNet20/50 and CIFAR10 / CIFAR100. This particular combination has showed specific effects in previous work, which unfortunately doesn’t always generalize. I am therefore somewhat skeptical to what degree the results generalize and would encourage the authors to systematically assess different architectures, sizes, and tasks. As is, the insight rest on relatively narrow experimental support. For example: the claim that going from hard to simple improves stability is evaluated only on CIFAR100 and CIFAR10, which are very entangled beyond ‘hard’ and ‘simple’. A claim that general requires more experimental evaluation.

The same goes for the comparison between language and vision models. Conventional vision models (like ResNet20s on CIFAR) and even relatively small and outdated BERT models operate in very different regimes (scale, parameter-data-ratio, training task, etc). The conclusion that language and vision models behave differently are not supported by a single skewed experiment like that. If anything, there may be insights into different behavior of different regimes. To compare different domains, one would have to compare models in the same regime varying the domain, e.g. large VITs pre-trained with data augmentation to recent LLMs, considering the respective scaling laws. In fact, the authors admit that in Section 5 lines 434.

**Essential References Not Discussed:**

N/A

**Experimental Designs Or Analyses:**

The experimental design is well done, except for the issues discussed above.

**Methods And Evaluation Criteria:**

Generally yes (modulo narrow datasets+architecture combinations discussed before). The framework is very well motivated and introduced. The only question I had is why the authors chose to use only loss barriers, l2 distance or perm distance? Why not evaluate behavioral similarlity (CKA, SVCCA, agreement, etc)? As the authors note, the same/different basins may be relevant for merging / ensembling models, for which it is essential to know whether or not models actually learn different behavior. I encourage the authors to include that in their framework as well, or to explain why they chose to not include it.

**Other Comments Or Suggestions:**

Typo in the paragraph heading in Section 5: "with with"

**Other Strengths And Weaknesses:**

## Strengths
- The paper is very well written, has a tight narrative and well motivated introduction.
- Relevant problem, understanding the sensitivity is fundamentally important, but also highly relevant for understanding the genesis of distinct phases, model merging, ensembling, pruning and also adversarial vulnerability of models.
- Well written related work, that clarified the submissions relation to existing work and identified the contribution.
- Extending Frankle et al. parent-child spawning experiments, the submission presents a more general framework to study the impact of some perturbation on a training trajectory.
- There are some very interesting results, particularly the attribution of stability to the weights (rather than direction), and suggesting the natural occurrence of permutations is not a likely phenomenon to explain trajectory divergence.

## Weaknesses
My main issues with the paper are already discussed above: the generality of claims that are not well supported experimentally, and the obmission of behavioral similarity metrics. Beyond that:
- There are several alignment methods that improve over git re-basin, e.g. Sinkhorn based [1] or learned [2].
- Figure 2 and 3 titles and caption could be improved. Left and middle figure have the same title. Why are Gaussian permutation barriers evaluated without accounting for permutations? Figure 3 - same as which of Figures 2?

[1] Pena et al., Re-basin via implicit Sinkhorn differentiation, CVPR 2023.
[2] Navon et al., Equivariant Deep Weight Space Alignment, ICML 2024

**Questions For Authors:**

I would be curious to learn why the authors chose their experimental setup as they have. Are these mostly computational constraints, or based on previous work?

Also, the submission motivates the work via two streams: optimization and work on early training phases. To me, that duality is lost in the course of the submission, and it is more of a generalization of existing work in the latter. I am curious what the authors see as the optimization perspective on their work.

**Relation To Broader Scientific Literature:**

The authors discuss the relation to previous work well throughout the paper, but explicitly in section 2.

**Theoretical Claims:**

N/A

---

> ### Author Rebuttal · Authors · 2025-04-01
>
> Thank you for your substantive review, to which we have replied below.
>
> ## F. Comparing Dissimilar Tasks and Scales
>
> > Experiments are performed mostly on relatively small vision models with relatively simple tasks. [...] To compare different domains, one would have to compare models in the same regime varying the domain,
>
> We agree with this assessment of our experimental scale, and analogous to figures 4-6, have added an experiment that fine-tunes ViTs from google/vit-base-patch16-224 on CIFAR-100: [https://imgur.com/a/AculLLH](https://imgur.com/a/AculLLH).
>
> We also note that the claims below (sections 4.3 and 5) are negative statements, for which a counterexample is sufficient:
>
> 1. More pre-training does not always improve fine-tuning stability (figures 4-6).
> 2. Barriers do not generally scale with $L^2$ divergence (figures 7-8).
> 3. Barriers and $L^2$ do not diverge exponentially proportional to perturbation scale (figure 9).
>
> In these cases, we do not strictly need to isolate the effects of task vs architecture or scaling.
>
> ## G. Fine-tuning Task Difficulty
>
> > the claim that going from hard to simple improves stability is evaluated only on CIFAR100 and CIFAR10, which are very entangled beyond ‘hard’ and ‘simple’.
>
> We agree that the claim “Pre-training on harder tasks improves stability” is weak. We attempted to make this claim more precise by relating stability to fine-tuning performance, but preliminary evidence in this direction is inconclusive. We have therefore replaced this claim with our new findings on CKA and ensembling (below).
>
> ## A., B. Functional Similarity And Model Diversity
> > Why not evaluate behavioral similarlity (CKA, SVCCA, agreement, etc)? As the authors note, the same/different basins may be relevant for merging / ensembling models, for which it is essential to know whether or not models actually learn different behavior.
>
> Reviewers *sufv* and *tDxT* have also raised this excellent point. Based on your feedback, we have run additional experiments that find:
> 1. Training instability (as evidenced by larger barriers) indeed correlates with functional dissimilarity (Angular CKA).
> 2. Perturbations to training can be used to improve ensemble performance, with more functional dissimilarity leading to better ensembles. This shows that instability increases model diversity.
>
> Please see our full rebuttal to Reviewer *sufv* for details and figures.
>
> Finally, to understand why we chose to focus on barriers as a measure of functional dissimilarity, please also see our rebuttal to Reviewer *HaCY* under **Non-Linear Connectivity**.
>
> ## Other Questions
>
> > There are several alignment methods that improve over git re-basin, e.g. Sinkhorn based [1] or learned [2].
>
> We note that weight matching (WM) is already able to greatly reduce barriers between *independently trained*  networks for our standard ResNet20-32 architecture [Ainsworth et al. 2023], whereas WM has little effect on the *identically spawned* networks of our experiments. The ineffectiveness of WM in our case suggests that other methods may not fare much better. Practically speaking, Peña et al. [2023] and Navon et al. [2024] are also quite costly (scaling poorly to larger models), as they must learn their permutations.
>
> > Figure 2 and 3 titles and caption could be improved. Left and middle figure have the same title. Why are Gaussian permutation barriers evaluated without accounting for permutations? Figure 3 - same as which of Figures 2?
>
> Thank you for the detailed corrections. We will revise the titles/captions, and add figures for permutation-aligned barriers for Gaussian perturbations into the appendix. Note that just like in batch perturbations, permutation does not reduce Gaussian-perturbed barriers.
>
> > I would be curious to learn why the authors chose their experimental setup as they have. Are these mostly computational constraints, or based on previous work?
>
> Indeed, our experimental setup is motivated by that of previous works [Frankle et al. 2020, Vlaar & Frankle 2022, Juneja et al. 2023, Ainsworth et al. 2023]. Having said that, evaluating stability for training (as opposed to fine-tuning) would require full runs which is costly when considering many replicates, perturbation times/magnitudes, and hyperparameter settings.
>
> > Also, the submission motivates the work via two streams: optimization and work on early training phases. To me, that duality is lost in the course of the submission, and it is more of a generalization of existing work in the latter. I am curious what the authors see as the optimization perspective on their work.
>
> We apologize for this confusion. Our intent was to disambiguate between our notion of instability (divergence of two trajectories that each independently converge to a solution) and that of work such as Wu et al. [2018] (tendency of a trajectory to fail to converge to certain solutions). We will de-emphasize the optimization perspective in the introduction to reflect its relevance to our work.

---

> > ### Comment · Reviewer_mmJd · 2025-04-02
> >
> > I would like to thank the authors for their rebuttal. While I appreciate computational burdens, I believe that further experimental evaluation beyond CIFAR would help greatly to assess how general or specific the findings are. I appreciate the author's reaction regarding fine-tuning task-difficulty. The addition of similarity scores, angular CCA or CKA, are a good step towards better understanding, and I encourage the authors to continue in this direction.
> > In light of the experimental evaluation and the other reviews, I will keep my score.

---

> > > ### Author Response · Authors · 2025-04-09
> > >
> > > Thank you for your thoughtful feedback. We fully agree that evaluating our findings across more diverse settings and larger models would strengthen their impact. However, given the computational scale needed to pre-train contemporary large models, we rely on publicly available intermediate checkpoints such as Multi-BERT to perform our large-scale fine-tuning experiments.
> > >
> > > To address your concerns while working within this constraint, we have expanded our experimental setting using AllenAI’s OLMo-1B large language model [1,2], which provides intermediates checkpoints throughout its ~740K step (3 trillion tokens) pretraining. We fine-tuned intermediate checkpoints of this model on GSM8K [3] for approximately 5,000 steps with a peak learning rate of 2e-5 and 10% warm-up. GSM8K contains grade school math problems and is used as a benchmark for many contemporary language models. We conducted our butterfly experiments with batch perturbations for fine-tuning stability at three pretraining checkpoints:
> > >
> > > 1. First available checkpoint (4B tokens)
> > > 2. Mid-way through pretraining (1.5T tokens)
> > > 3. Final checkpoint (3T tokens)
> > >
> > > Our results are plotted here: [https://imgur.com/a/KQ4kz4D](https://imgur.com/a/KQ4kz4D).
> > > This experiment strongly corroborates our Multi-BERT findings (Fig. 6 in the paper): namely, we observe that pre-training for longer actually reduces stability to fine-tuning. Moreover, our insights regarding perturbation time and scale, whereby earlier and larger perturbations result in higher barriers, remain unchanged in the larger OLMo setting.
> > >
> > > Our findings highlight an important consideration for the field. Typically fine-tuning is performed only on the latest checkpoints, which are in turn used for model merging or MoErging (mixture-of-experts method) [4]. Our results 	imply that depending on the fine-tuning task, earlier checkpoints may be better suited for transfer learning and model merging, and that the optimal pre-training duration could be related to the stability of a given checkpoint with respect to the fine-tuning task. Our work also highlights the value of making intermediate checkpoints more widely available for research and practical purposes.
> > >
> > > Regarding similarity scores, we greatly appreciated your feedback and we will adjust our paper accordingly. While computational and time constraints prevent us from providing comprehensive CKA analyses for transformer models in time to add to this comment, we will include these metrics for select subsets of our fine-tuning experiments in the final version of our paper.
> > >
> > > [1] https://huggingface.co/allenai/OLMo-1B-hf
> > >
> > > [2] Groeneveld D., et al. OLMo: Accelerating the Science of Language Models. 2024.
> > >
> > > [3] Cobbe, K., et al. Training Verifiers to Solve Math Word Problems, 2021.
> > >
> > > [4] Yadav P., et al. A Survey on Model MoErging: Recycling and Routing Among Specialized Experts for Collaborative Learning, 2024.

---

### Official Review · Reviewer_tDxT · 2025-03-09

**Overall Recommendation:** 3

**Summary:**

In this work, the authors study how the sensitivity of training trajectories in neural network training depends changes with the distance to the initialization. The authors characterize the sensitivity through the divergene of training trajectories, which is measured in $L_2$ distance, loss barrieres, defined as $\sup_{\alpha \in (0,1)} \ell(\alpha \theta_T + (1-\alpha) \theta'_T) - \alpha \ell(\theta_T) - (1-\alpha) \ell(\theta'_T)$, and barriers module permutation.

Different from previous work, the authors study the training stability by removing the noise and studying deterministic training with controlled pertubations. The authors also compare the stability of loss landscape induced by different choices of optimizers and hyperparameters.

Their main findings include:
- a small pertubation at a single iteration, which is smaller than training noise, is sufficient to cause the divergence of two otherwise identically and trained networks.
- stability during the early phase of training can be improved for wider or shallower networks, increasing the batch size, learning rate warm-up and weight decay.
- pre-trained networks are multiple orders of magnitudes more stable than randomly initialized networks.

**Claims And Evidence:**

Given the metrics that the authors have chosen to measure divergence and stability of training runs, I think the claims of the authors are well supported by the evidence provided, for instance in Figure 2 (early pertubation has a much larger effect than later pertubation), Figure 3 (modifying training hyperparameters can reduce the butterfly effect) or Figure 4 (pre-trained networks are much more stable than randomly-initialized networks).

**Essential References Not Discussed:**

There is some related work on mode connectivity, which have not discussed in this work, including Draxler et al. [2018] on "Essentially No Barriers in Neural Network Energy Landscape" and Garipov et al. [2018] on "Loss Surfaces, Mode Connectivity, and Fast Ensembling of DNNs".

Both work show that minima reached from different initializations can be connected through simple, non-linear curves. Can the authors discuss how this paper relates to these works?

**Experimental Designs Or Analyses:**

I checked the training details described in Appendix A and experiment details in Appendix B, which I believe is fine. Also the computation of the loss barriers seems reasonable to me. However I am not familier with the matching algorithm from Ainsworth et al. [2023] to find permutation of neurons $P$ that approximately minimizes the $L_2$ distance between two networks' weights.

**Methods And Evaluation Criteria:**

I think that the evaluation criteria are reasonable overall, by considering both the loss barrier with and without permutation, the authors try to exclude the fact that training instability merely causes permutations between networks (although it cannot excluded, as the authors state, that they simply failed to find barrier-reducing permutations).

Also the $L_2$ distance is an intuitive metric to quantify the divergence between two runs. In fact, it seems even more natural to me to use this metric (or after applying $L_2$-minimizing permutations). What is the reason for the authors not to show these figures and can they provide them during the rebutal period?

Given the fact that there is some work, e.g. Draxler et al. [2018] or Garipov et al. [2018], which showed that the optima in neural networks from different initializations can be connected through (simple) non-linear curves, I wonder how much it makes sense to look only at the barriers along the linear path between the weights. It would be interesting to understand whether one can observe a similar trend if one computes the barrier along a Bezier curve. See for instance Fig. 1 in Garipov et al. [2018].

**Other Comments Or Suggestions:**

It would be helpful to show Figures 2-6 and Figure 9 with the y-axis in log-scale to distinguish the lines even better.

**Other Strengths And Weaknesses:**

+ + I think the main strength and novelty of this work is to isolate the effect of noise in training and really just comparing (deterministic) training runs with controlled injected pertubations. This highlights the sensitivity of the network to slight variations.

- - I think the main weakness of this paper is that it is still unclear to me in what way the diverged solutions are actually "different" from each other, since they achieve comparable training and test accuracy. For instance, it could be interesting to understand whether all pertubations still converge to the same manifold and how different pertubations for different points in time and magnitude of pertubations are distributed to each other (in terms of $L_2$ distance). The authors also mentioned that it is desired in ensembling methods to combine diverse solutions, but I am not sure what insights this work provides in this regard. I think my main concern with this work is that I am not sure what a reasonable way is to connect divergence to diversity of solutions.
Can the authors perhaps comment on this?

**Questions For Authors:**

I was first suprised that there is one point for $t=0.0 \%$ in Figure 2 where the barrier is actually higher after permutation than without. Upon closer look I realized that the permutation is minimizing the $L_2$ distance between two networks.
Can the authors therefore provide similar Figures as the ones in Figure 2 or Figure 3, where the error barrier is measured in terms of $L_2$ divergence? It would particular interesting for me to see, how much the permutation of neurons $P$ can reduce the $L_2$ distance between two networks.

What are the conclusions on network training that you draw from the results of this paper?

Although I gave a rather low score, I am open to adjust my score if the authors address my comments.

**Relation To Broader Scientific Literature:**

This paper is strongly related to work on linear mode connectivity, which has been studied extensively, for instance in Frankle et al. [2020a;b], Singh et al. [2020.]. Bachmann et al. [2023] study different factors which lead to linear mode connectivity (or lack thereof), including the network architecture, training setup and the dataset, while Entezari et al. [2022] pose the conjecture that by taking into account permutation invariance, solutions trained with SGD will likely be in the same basin.

This paper follows the experimental setup as in Frankle et al. (2020a), but eliminate the effect of training noise by fixing it for different runs and isolate the effect of pertubations.

**Theoretical Claims:**

This paper does not contain any proof. The derivation of the lineared approximation for $L_2$ divergence in Appendix C looks correct to me.

---

> ### Author Rebuttal · Authors · 2025-04-01
>
> Thank you for the detailed review, which we address by topic below.
>
> ## E. $L^2$ Divergence
>
> > Also the $L^2$ distance is an intuitive metric to quantify the divergence between two runs. [...] Can the authors therefore provide similar Figures as the ones in Figure 2 or Figure 3, where the error barrier is measured in terms of divergence? It would particular interesting for me to see, how much the permutation of neurons $P$ can reduce the $L^2$ distance between two networks.
>
> Apologies for the omission. We have plotted $L^2$ divergence in place of barriers here and will add them to the appendix for all figures: [https://imgur.com/a/BKMQHQk](https://imgur.com/a/BKMQHQk).
>
> We do not prioritize $L^2$ divergence as it is not as indicative of functional similarity as barriers. $L^2$ is sensitive to optimization choices (e.g. weight decay), and network scale symmetries. Indeed, we observe that:
>
> - The relationship between $L^2$ and barriers differs depending on hyperparameter settings for our vision experiments (section 5, figure 7 colors).
> - $L^2$ and barriers are unrelated for our language experiments (section 5, figure 8).
>
> Regarding weight matching (WM), Ito et al. [2024] finds:
> > permutations found by WM do not significantly reduce the distance between the two
> models [...] WM satisfies LMC by aligning the directions of singular vectors with large singular values in each layer’s weights.
>
> This highlights another lack of correspondence between $L^2$ and barriers.
>
> - Ito, A., et al. (2024). Analysis of linear mode connectivity via permutation-based weight matching
>
>
> ## C. Non-Linear Connectivity
>
> > Given the fact that there is some work, e.g. Draxler et al. [2018] or Garipov et al. [2018], which showed that the optima in neural networks from different initializations can be connected through (simple) non-linear curves, I wonder how much it makes sense to look only at the barriers along the linear path between the weights. [..] Both work show that minima reached from different initializations can be connected through simple, non-linear curves. Can the authors discuss how this paper relates to these works?
>
> This is an important point also raised by Reviewer *HaCY*. Please see our rebuttal to Reviewer *HaCY* under **C. Non-Linear Connectivity**, where we address this issue.
>
> ## A., B. Functional Similarity And Model Diversity
>
> > I think the main weakness of this paper is that it is still unclear to me in what way the diverged solutions are actually "different" from each other, [...] For instance, it could be interesting to understand whether all pertubations still converge to the same manifold and how different pertubations for different points in time and magnitude of pertubations are distributed to each other (in terms ofdistance).
>
> > The authors also mentioned that it is desired in ensembling methods to combine diverse solutions, [...] I think my main concern with this work is that I am not sure what a reasonable way is to connect divergence to diversity of solutions. Can the authors perhaps comment on this?
>
> These are important observations echoed by Reviewers *sufv* and *mmJd*. Based on your feedback, we have conducted additional experiments using CKA and ensembling to measure model diversity. Please see our full rebuttal to Reviewer *sufv* for details.
>
> Regarding the first quoted passage, could you clarify what you mean by “converge to the same manifold”? This may be related to the section **C. Non-Linear Connectivity**.
>
> ## Other Questions
>
> > It would be helpful to show Figures 2-6 and Figure 9 with the y-axis in log-scale to distinguish the lines even better.
>
> We have updated these figures accordingly and will add them to the text:
> - Figures 2 and 3 [https://imgur.com/a/fig2-3-jND4s6E](https://imgur.com/a/fig2-3-jND4s6E).
> - Figures 4, 5, and 6 [https://imgur.com/a/hsJyGhZ](https://imgur.com/a/hsJyGhZ). We provide tablbes in place of Figure 4 as barriers are 0 for $\sigma<0.01$.
> - Figure 9 [https://imgur.com/a/jUXmcZQ](https://imgur.com/a/jUXmcZQ).
>
> > What are the conclusions on network training that you draw from the results of this paper?
>
> Our main conclusion is that neural network training is unstable near initialization even without randomness, meaning that interventions to reduce SGD noise (larger batch size, reduced learning rates, longer warmup periods) cannot eliminate this instability. Similar work has found that the trainability of neural networks can be unstable to small hyperparameter changes [Sohl-Dickstein 2024].
>
> Conversely, pre-trained networks can still be fine-tuned into different modes (and thus, diverse solutions) with sufficiently large perturbations. Our findings suggest ways to manage model diversity among multiple training runs, such as by adjusting pre-training length, changing SGD noise magnitude via hyperparameters, or by directly perturbing networks.
>
> We will add this conclusion into the text.
>
> - Sohl-Dickstein, J. (2024). The boundary of neural network trainability is fractal.

---

### Official Review · Reviewer_HaCY · 2025-03-12

**Overall Recommendation:** 3

**Summary:**

This paper studies the impact of perturbation on SGD in an empirical way. The authors focus on the condition under which the perturbation during training can lead to convergence to another basin. The authors prepared perturbation in different directions of various scales that is applied to training at different moments. The same experiment is performed for both random initialization and transfer learning. The notable findings are:
* Stability increases as the training goes on;
* Stability can be improved by hyperparameter tuning;
* pre-trained models are more stable;
* Long pre-training reduces stability when fine-tuning language models.

**Claims And Evidence:**

All of the claims are supported by numerical evidences. The authors provide statistically significant results in controlled settings and the experiment themselves are solid. However, the interpretation of the results may be less so. I think characterizing the basin by a linearly connected low-loss region in the loss landscape is a little problematic. (I will leave the details in theoretical claims).

**Essential References Not Discussed:**

To my knowledge there are no essential references not discussed.

**Experimental Designs Or Analyses:**

I am not very familiar with fine-tuning language models, so I cannot check the experimental designs in detail in this regard. I don’t find notable problems in experiments in other cases.

**Methods And Evaluation Criteria:**

Thus authors use image classification and language model training problems to study the stability. These are reasonable choices and they are good enough to support the contents of this paper, but they don’t cover all possible cases for learning (prediction, RL, PINN, …).

**Other Comments Or Suggestions:**

I have no other comments.

**Other Strengths And Weaknesses:**

The authors investigate a lot of potential causes for instability. These potential causes are well isolated in the experiments, and the experimental facts are recorded well. This paper can serve as a nice starting point for crafting theories about stability.

In this work, the authors freeze the noise in SGD, and the injected noise is the main object of study for stability analysis. In reality, the noise of SGD may be much larger than the perturbations studied in this paper and overshadow them. Thus, I don't think the stability the authors study is very relevant.

**Questions For Authors:**

I wonder if the authors know how to tell whether $\theta_T$ and $\theta’_T$ are connected by continuous symmetry or not.

I wonder if the authors could quantify how important the perturbation studied in this paper is compared to the SGD noise.

In the numerical experiments, I have the impression that the amplitude of the perturbation can be much larger than rounding error. Where could these perturbations possibly come from?

**Relation To Broader Scientific Literature:**

I think this paper provides an alternative point of view to existing works. As far as I know the stability works usually study the problem that at which local minimum the SGD stays. The authors instead study whether SGD converges to the same local minimum regardless of the characteristic of the local minimum. Also, I am not aware of previous works that freezes SGD noise and introduce another source of noise.

**Theoretical Claims:**

This paper does not involve theoretical proofs.

I find the reasoning a little problematic though. The authors characterize the basin by a linearly connected low-loss region in the loss landscape. I don’t think this is always true. There is symmetry (not limited to permutation) in the loss function (for example, as discussed in https://arxiv.org/pdf/2309.16932). Due to the continuous symmetries, there are equivalent local minima that should be considered the same basin and  are not linearly connected. A naive example would be a loss function for parameters $(u, v)$ and data point $(x, y)$: $(uvx - y)^2$. The minima are located on the curve $uv = y/x$, and these solutions provide the same training and testing performance. However, these solutions are not considered to be in the same basin according to the authors. Thus, I think the authors’ way of modeling the basin does not take into account the complexity in the landscape.

---

> ### Author Rebuttal · Authors · 2025-04-01
>
> Thank you for your thoughtful review. We have addressed the quoted comments and questions point-by-point below.
>
> ## C. Non-Linear Connectivity
>
> > Due to the continuous symmetries, there are equivalent local minima that should be considered the same basin and are not linearly connected. [...] However, these solutions are not considered to be in the same basin according to the authors. [...] I wonder if the authors know how to tell whether $\theta_T$ and $\theta’_T$ are connected by continuous symmetry or not.
>
> Thank you for raising this crucial and subtle distinction, which is also suggested by Reviewer *tDxT*.
>
> Our definition of “loss basin” (which we refer to here as “LMC basin”) in section 2 (*Linear mode connectivity*) reflects when the loss landscape is locally convex, and is drawn from Neyshabur et al. [2020] and Frankle et al. [2020]. Although this perspective is more restrictive than a general notion of “mode connectivity” [Draxler et al. 2018, Garipov et al. 2018], ours is both theoretically and practically relevant:
> 1. Despite being a non-convex problem, neural network training has many connections with convex optimization. LMC basins describe approximately convex regions of the loss landscape.
> 2. Classical methods for determining the stability of dynamical systems take a linear or quadratic approximation, which holds precisely in LMC basins.
> 3. Model merging applications require networks to share a LMC basin, as shown by permutation alignment research [Singh & Jaggi 2020, Entezari et al. 2022, Ainsworth et al. 2023].
>
> Prior work, both theoretical [Simsek et al. 2021, Lin et al. 2024] and empirical [Draxler et al. 2018, Garipov et al. 2018, Sonthalia et al. 2024], also points to non-linear connectivity being a trivial property of neural network minima in general. If this is the case, then knowing if $\theta_T$ and $\theta’_T$ are continuously connected would be rather uninformative.
>
> We apologize for not making our definition of “loss basin” and our justifications for the choice clear, and will update the above points in the text.
>
> - Simsek, B., et al. (2021). Geometry of the Loss Landscape in Overparameterized Neural Networks: Symmetries and Invariances.
> - Lin, Z.,et al. (2024). Exploring neural network landscapes: Star-shaped and geodesic connectivity.
> - Sonthalia, A., et al. (2024). Do Deep Neural Network Solutions Form a Star Domain?.
>
> ## D. Correspondence With SGD Noise
>
> > In this work, the authors freeze the noise in SGD, and the injected noise is the main object of study for stability analysis. In reality, the noise of SGD may be much larger than the perturbations studied in this paper and overshadow them. Thus, I don't think the stability the authors study is very relevant. [...] I wonder if the authors could quantify how important the perturbation studied in this paper is compared to the SGD noise.
>
> We agree that we have not sufficiently established the relative magnitude of our perturbations vs SGD noise. To address this, we replicate the parent-child spawning experiment of Frankle et al. [2020] (Frankle baseline) and show that our batch perturbations are a lower bound on the Frankle baseline’s barriers [https://imgur.com/a/frankle-ThKay8T](https://imgur.com/a/frankle-ThKay8T). In this comparison, we scale batch perturbations to the expected magnitude of SGD noise at the perturbation time $t$: making batch perturbation equivalent to taking only one step at time $t$ with different SGD noise, as opposed to using different SGD noise from $t$ onwards in the Frankle baseline.
>
> We also tested even smaller perturbations by perturbing a fraction of weights at the smallest perturb scale we used in our main experiments ($10^{-4}$).
> We find that as little as a **single (!)** perturbed weight (at a fraction of $10^{-6}$) causes barriers at initialization [https://imgur.com/a/AvfP8Mh](https://imgur.com/a/AvfP8Mh), which is well below the scale of noise from sources such as hardware indeterminacy.
>
> The significance of our approach is that, as per section 3.1, the exponential convergence or divergence of a deterministic dynamical system dominates the diffusion effects of SGD noise [Wu et al. 2018]. This means that, in the dynamical systems model, training trajectories will diverge as long as they are unstable to noise *once*, whereas in the diffusion model, divergence depends on noise persisting throughout training.
>
> ## Other Questions
>
> > In the numerical experiments, I have the impression that the amplitude of the perturbation can be much larger than rounding error. Where could these perturbations possibly come from?
>
> Sources of perturbation in regular training could include SGD noise (batch order, data augmentation, hardware indeterminism) as well as model pruning or quantization.  We do not target a specific source so as to keep our experimental findings general.

---

### Official Review · Reviewer_sufv · 2025-03-15

**Overall Recommendation:** 4

**Summary:**

The paper studies the impact of applying isolated perturbations to model parameters during different training and fine-tuning phases. The analysis provides insights into model training stability using three quantities measuring parameter similarity or functional similarity. The findings suggest that models become more stable during training, and the effect is even more evident for pretrained models. Interestingly, analysis shows that the longer training in the case of language tasks hurts the model's stability.

**Claims And Evidence:**

All the claims are supported by clear empirical evidence.

**Essential References Not Discussed:**

None

**Experimental Designs Or Analyses:**

The experiments are clearly explained to the reader and meticulously designed to precisely measure the effect of perturbations.

**Methods And Evaluation Criteria:**

The methods (models and benchmarks) cover various popular settings, making the study interesting for both image classification and NLP communities.

The measures used by the authors to estimate the similarities between models are well-established methods within this area. However, I'm wondering how other similarity measures differ from the perspective presented by the authors, e.g., CCA or CKA index, which could be an interesting add-on to the current work.

**Other Comments Or Suggestions:**

None

**Other Strengths And Weaknesses:**

As mentioned earlier, it would be interesting to broaden the scope of methods for computing similarity between two models. Measures such as CCA, CKA could be useful in this regard and could offer a deeper understanding of the phenomenon.

**Questions For Authors:**

None

**Relation To Broader Scientific Literature:**

The work is well positioned within the related works. My current understanding is that the work could be an important first step towards designing precise training manipulations, which in turn could be used to increase the model's performance or robustness to data shifts by model merging techniques. It would be interesting to continue this line of work and explore questions like: How can optimally perturb a model (and at which point) increase the final model's performance of the averaged model? Would that perturbation be universal across different merging strategies or should each strategy have its own method of perturbation?

**Theoretical Claims:**

The paper does not contain any formal statements.

---

> ### Author Rebuttal · Authors · 2025-04-01
>
> Thank you for your insightful review. Please find our replies below for the quoted points.
>
> ## A. Other Similarity Measures
>
> > I'm wondering how other similarity measures differ from the perspective presented by the authors, e.g., CCA or CKA index, which could be an interesting add-on to the current work.
>
> This is an excellent idea and was also recommended by Reviewer *mmJd*.
> We prioritized barriers in our original analysis in order to determine when networks are in the same locally convex loss basin. For our reasoning about why this is useful, see the discussion of the advantages of linear mode connectivity in our rebuttal to Reviewer *HaCY*, under **C. Non-Linear Connectivity**.
>
> Representational similarity measures (e.g. CKA) are a more generic way to measure functional similarity that is not sensitive to mechanistic differences between neural network weights, such as barriers that prevent effective model merging. Nevertheless, we agree that measuring similarity in a more general sense will better illuminate how perturbed training trajectories differ.
>
> Accordingly, we have now conducted the following experiment on our standard setting (figure 2):
>
> 1. We compute output dissimilarity (in $L^2$ between logits, or % of disagreeing classifications) and Angular CKA [Williams et al. 2021] between pairs of networks in our experiments: [https://imgur.com/a/zOBlg5X](https://imgur.com/a/zOBlg5X).
> 2. As expected, measures of functional dissimilarity increase as barriers increase.
> 3. Functional similarity becomes more sensitive to batch perturbation later in training.
>
> Note that Angular CKA ranges from 0 (perfectly similar) to $\pi$ (perfectly dissimilar), with $\pi/2$ indicating no correlation. We plot the largest (most dissimilar) CKA value over all residual block outputs, which are computed over 10000 examples using software from [Lange et al. 2023].
>
> We will replicate the above experiment for our fine-tuning settings for inclusion in the text.
>
> - Williams, A. H.,et al. (2021). Generalized shape metrics on neural representations.
> - Lange, R. D., et al. (2023). Deep networks as paths on the manifold of neural representations.
>
> ## B. Model Diversity
>
> > How can optimally perturb a model (and at which point) increase the final model's performance of the averaged model? Would that perturbation be universal across different merging strategies or should each strategy have its own method of perturbation?
>
> This is a very interesting and novel application which we did not previously consider.
> We have interpreted your point as the hypothesis that *deliberately perturbing models increases model diversity, leading to improved ensembling performance*. We test this hypothesis as follows:
>
> 1. We take our measurements from above, and plot Angular CKA (x-axis) against ensemble performance: [https://imgur.com/a/ah7kCfg](https://imgur.com/a/ah7kCfg).
> 2. We find that the ensembles indeed perform better on more dissimilar model pairs, which is in line with the proposed hypothesis.
> 3. Note that, as shown by barriers increasing with Angular CKA, model averaging performs worse for more dissimilar model pairs.
>
> By ensembling, we mean that in each of our spawn-and-perturb experiments, we evaluate the average of the output logits for the unchanged and perturbed networks after training.
> While Gaussian perturbations appear to have a more consistent relationship between functional dissimilarity and ensemble performance, we leave a full exploration of which perturbations are best suited for different merging strategies to future work.

---

> > ### Comment · Reviewer_sufv · 2025-04-02
> >
> > Thank the authors for addressing my concerns and providing additional experiments. I still believe that the findings from this work could serve as a starting point for finding optimal perturbations that increase ensemble performance and deserve being accepted.

---

> > > ### Author Response · Authors · 2025-04-09
> > >
> > > Thank you for the positive assessment of our work. We will include additional CKA analyses such as for fine-tuning transformer models in the final version of our paper. Additionally, we have completed a larger-scale experiment as suggested by Reviewer *mmJd*, the results of which we have appended here for convenience:
> > >
> > > ## Large-scale experiments
> > >
> > > We have expanded our experimental setting using AllenAI’s OLMo-1B large language model [1,2], which provides intermediates checkpoints throughout its ~740K step (3 trillion tokens) pretraining. We fine-tuned intermediate checkpoints of this model on GSM8K [3] for approximately 5,000 steps with a peak learning rate of 2e-5 and 10% warm-up. GSM8K contains grade school math problems and is used as a benchmark for many contemporary language models. We conducted our butterfly experiments with batch perturbations for fine-tuning stability at three pretraining checkpoints:
> > >
> > > 1. First available checkpoint (4B tokens)
> > > 2. Mid-way through pretraining (1.5T tokens)
> > > 3. Final checkpoint (3T tokens)
> > >
> > > Our results are plotted here: [https://imgur.com/a/KQ4kz4D](https://imgur.com/a/KQ4kz4D).
> > > This experiment strongly corroborates our Multi-BERT findings (Fig. 6 in the paper): namely, we observe that pre-training for longer actually reduces stability to fine-tuning. Moreover, our insights regarding perturbation time and scale, whereby earlier and larger perturbations result in higher barriers, remain unchanged in the larger OLMo setting.
> > >
> > > Our findings highlight an important consideration for the field. Typically fine-tuning is performed only on the latest checkpoints, which are in turn used for model merging or MoErging (mixture-of-experts method) [4]. Our results 	imply that depending on the fine-tuning task, earlier checkpoints may be better suited for transfer learning and model merging, and that the optimal pre-training duration could be related to the stability of a given checkpoint with respect to the fine-tuning task. Our work also highlights the value of making intermediate checkpoints more widely available for research and practical purposes.
> > >
> > > [1] https://huggingface.co/allenai/OLMo-1B-hf
> > >
> > > [2] Groeneveld D., et al. OLMo: Accelerating the Science of Language Models. 2024.
> > >
> > > [3] Cobbe, K., et al. Training Verifiers to Solve Math Word Problems, 2021.
> > >
> > > [4] Yadav P., et al. A Survey on Model MoErging: Recycling and Routing Among Specialized Experts for Collaborative Learning, 2024.

---

### Decision · Program_Chairs · 2025-05-01

**Decision:**

Accept (poster)

**Comment:**

The paper systematically studies the chaotic nature of neural network training i.e., little changes to initial conditions smaller than training noise, leads to highly divergent later trajectories.

All reviewers unanimously appreciated the paper.
- Most reviewers appreciated the breadth of models studied (`sufv`)
- and the fact that both image and language models are studied (`sufv`, `HacY`).
- There is agreement that the experiments are rigorous and support the claims (`HacY`: "claims are supported by numerical evidences", "statistically significant results in controlled settings", `tDXT`: "the claims of the authors are well supported by the evidence provided").
- The reviewers also find the results interesting and an advance on what is studied in this area (`mmJD`).

I believe that the main concerns were addressed well by the authors through additional results/arguments:
1. Adding other similarity scores (`sufv`, `mmJD`, `tDxT `)
3. Models/tasks being outdated/small/simple (`mmJD`)
3. Why study linear mode connectivity (`HacY`, `tDXT`, ) -- the linearity of mode connectivity only makes it a more interesting phenomenon to study.
4. Noise studied is smaller than that of training noise (`HacY`) -- like above, I believe that this only makes the instabilities a more surprising object worth a study.

_Footnote that doesn't affect the decision:_ It would be worth relating the curious findings here to the one in [1] where a student model initialized _very_ close the teacher ends up deviating far from the teacher (see their Fig 5 and discussion below).

"Random Teachers are Good Teachers", Sarnthein et al., ICML 2023, https://arxiv.org/abs/2302.12091